# Understanding Contrastive Learning via Distributionally Robust Optimization

**Junkang Wu**[1]  **Jiawei Chen**[2*] **Jiancan Wu**[1]  **Wentao Shi**[1]  **Xiang Wang**[1†] **Xiangnan He**[1*]
[1]University of Science and Technology of China
[2]Zhejiang University
{jkwu0909, wujcan, xiangwang1223, xiangnanhe}@gmail.com,
sleepyhunt@zju.edu.cn, shiwentao123@mail.ustc.edu.cn

## Abstract

This study reveals the inherent tolerance of contrastive learning (CL) towards sampling bias, wherein negative samples may encompass similar semantics (*e.g.,* labels). However, existing theories fall short in providing explanations for this phenomenon. We bridge this research gap by analyzing CL through the lens of distributionally robust optimization (DRO), yielding several key insights: (1) CL essentially conducts DRO over the negative sampling distribution, thus enabling robust performance across a variety of potential distributions and demonstrating robustness to sampling bias; (2) The design of the temperature $\tau$ is not merely heuristic but acts as a Lagrange Coefficient, regulating the size of the potential distribution set; (3) A theoretical connection is established between DRO and mutual information, thus presenting fresh evidence for "InfoNCE as an estimate of MI" and a new estimation approach for $\phi$-divergence-based generalized mutual information. We also identify CL's potential shortcomings, including over-conservatism and sensitivity to outliers, and introduce a novel Adjusted InfoNCE loss (ADNCE) to mitigate these issues. It refines potential distribution, improving performance and accelerating convergence. Extensive experiments on various domains (image, sentence, and graphs) validate the effectiveness of the proposal. The code is available at https://github.com/junkangwu/ADNCE.

## 1 Introduction

Recently, contrastive learning (CL) [1; 2; 3] has emerged as one of the most prominent self-supervised methods, due to its empirical success in computer vision [4; 5; 6; 7; 8], natural language processing [9; 10] and graph [11; 12]. The core idea is to learn representations that draw positive samples (*e.g.,* augmented data from the same image) nearby and push away negative samples (*e.g.,* augmented data from different images). By leveraging this intuitive concept, unsupervised learning has even begun to challenge supervised learning.

Despite its effectiveness, recent study [13] has expressed concerns about sampling bias in CL. Negative samples are often drawn uniformly from the training data, potentially sharing similar semantics (*e.g.,* labels). Blindly separating these similar instances could result in a performance decline. To address sampling bias, existing research has proposed innovative solutions that either approximate the optimal distribution of negative instances [13; 14] or integrate an auxiliary detection module to identify false negative samples [15]. Nevertheless, our empirical analyses challenge the prevailing understanding of CL. Our findings indicate that CL inherently exhibits robustness towards

---

*Jiawei Chen and Xiangnan He are the corresponding authors.

†Xiang Wang is also affiliated with Institute of Artificial Intelligence, Institute of Dataspace, Hefei Comprehensive National Science Center.

37th Conference on Neural Information Processing Systems (NeurIPS 2023).

sampling bias and, by tuning the temperature $\tau$, can even achieve comparable performance with representative methods for addressing sampling bias (*e.g.,* DCL [13] and HCL [14]). This revelation prompts two intriguing questions: *1) Why does CL display tolerance to sampling bias? and 2) How can we thoroughly comprehend the role of the temperature $\tau$?* Despite recent analytical work on CL from various perspectives [13; 14; 16], including mutual information and hard-mining, these studies fail to provide a satisfactory explanation for our observations.

To address this research gap, we investigate CL from the standpoint of Distributionally Robust Optimization (DRO) [17; 18; 19; 20; 21; 22]. DRO refers to a learning algorithm that seeks to minimize the worst-case loss over a set of possible distributions. Through rigorous theoretical analysis, we prove that CL essentially performs DRO optimization over a collection of negative sampling distributions that surround the uniform distribution, constrained by a KL-divergence-based measure (we term the DRO-type equivalent expression of CL as CL-DRO). By enabling CL to perform well across various potential distributions, DRO equips it with the capacity to alleviate sampling bias. Furthermore, our findings emphasize that the design of $\tau$ is not merely heuristic but acts as a Lagrange Coefficient, regulating the size of the potential distribution set. By integrating the principles of DRO, we can also gain insights into the key properties of CL such as hard-mining, and investigate novel attributes such as variance control.

Furthermore, we delve into the properties of CL-DRO, transcending the boundary of KL divergence, but exploring a broader family of distribution divergence metrics — $\phi$-divergence. It exhibits greater flexibility and potentially superior characteristics compared to KL-divergence. Interestingly, we establish the equivalence between CL-DRO under any $\phi$-divergence constraints and the corresponding tight variational representation of $\phi$-divergence [23]. This discovery provides direct theoretical support for the assertion that "*InfoNCE is an estimate of mutual information (MI)*", without requiring redundant approximation as before work [24; 3]. Additionally, it provides a novel perspective for estimating arbitrary $\phi$-divergence, consequently enabling the estimation of $\phi$-divergence-based mutual information.

In addition, we discover some limitations in CL through the lens of DRO. One is that the DRO paradigm is too conservative, focusing on worst-case distribution with giving unadvisedly over-large weights on hard negative samples. Secondly, the presence of outliers can significantly impact DRO [25], resulting in both a drop in performance and training instability. In fact, both challenges stem from the unreasonably worst-case distribution weights assigned to negative samples. To address this weakness, we propose **AD**justed Info**NCE** (ADNCE) that reshapes the worst-case distribution without incurring additional computational overhead, allowing us to explore the potentially better negative distribution. Empirical studies show that our method can be applied to various domains (such as images, sentences, and graphs), and yield superior performance with faster convergence.

## 2 Related Work

**Contrastive Learning.** With numerous CL methodologies (*e.g.,* SimCLR [4], Moco [6], BYOL [26], SwAV [27], Barlow Twins [8]) having been proposed and demonstrated the ability to produce high-quality representations for downstream tasks, theoretical research in this field has begun to emerge. Wang et al. [28] revealed the hard-mining property and Liu et al. [29] emphasized its robustness against dataset imbalance. Meanwhile, Xue et al. [30] showed that representations learned by CL provably boosts robustness against noisy labels, and Ko et al. [31] investigated the relationship between CL and neighborhood component analysis. More closely related to our work is the result of Tian et al. [16] that analyzed CL from minmax formulation.

**Distributionally Robust Optimization.** In contrast to conventional robust optimization approaches [32; 33; 34], DRO aims to solve the distribution shift problem by optimizing the worst-case error in a pre-defined uncertainty set, which is often constrained by $\phi$-divergence [20; 17], Wasserstein distance [21; 22; 19; 18] and shape constraints [35; 36; 37]. Meanwhile, Michel et al. [38; 39] parametrized the uncertainty set, allowing more flexibility in the choice of its architecture. Zhai et al. [25] focused on issues related to the existing DRO, such as sensitivity to outliers, which is different from the aforementioned research.

**Mutual Information.** The MI between two random variables, $X$ and $Y$, measures the amount of information shared between them. To estimate MI, the Donsker-Varadhan target [40] expresses the lower bound of the Kullback-Leibler (KL) divergence, with MINE [24] being its parameterized

version. However, according to studies [41; 42], the MINE estimator may suffer from high variance. To address this issue, InfoNCE [3] extends the unnormalized bounds to depend on multiple samples, thereby eliminating the constraints of both MINE's reliance on conditional and marginal densities. Another line of MI focuses on boundedness and low variance. Towards this end, RPC [43] introduced relative parameters to regularize the objective and FLO [44] leveraged Fenchel-Legendre optimization. Nevertheless, they are still hard to explain *how MI optimization elucidates the properties of CL and facilitates effective CL in practice.*

## 3 Contrastive learning as Distributionally Robust Optimization

In this section, we firstly uncover the tolerance of CL towards sampling bias. Then we analyze CL and $\tau$ from the perspective of DRO. Finally, we empirically verify the above findings.

**Notation.** Contrastive Learning is conducted on two random variables $X, Y$ over a feature space $\mathcal{A}$. Suppose we have original samples $x$, which obey the distribution $P_X$, positive samples drawn from the distribution $(x, y^+) \sim P_{Y|X}$, and negative samples selected from the distribution $(x, y) \sim P_Y$. The goal of CL is to learn an embedding $g_\theta : \mathcal{A} \to \mathbb{R}^d$ that maps an observation (*i.e., $x, y, y^+$*) into a hypersphere. Let $f(g_\theta(x), g_\theta(y))$ be the similarity between two observations $x$ and $y$ on the embedding space. CL draws positive samples closer (*i.e., $f(g_\theta(x), g_\theta(y^+)) \uparrow$ for $(x, y^+) \sim P_{XY}$*) and pushes the negative samples away (*i.e., $f(g_\theta(x), g_\theta(y)) \downarrow$ for $(x, y) \sim P_X P_Y$*). For brevity, in this work, we simply shorten the notations $P_{Y|X}$ as $P_0$, and $P_Y$ as $Q_0$. Also, we primarily focus on InfoNCE loss $\mathcal{L}_{\text{InfoNCE}}$ as a representative for analyses, while other CL's loss functions exhibit similar properties. InfoNCE [3] can be written as follow:

$$\mathcal{L}_{\text{InfoNCE}} = -\mathbb{E}_{P_X} \left[ \mathbb{E}_{P_0}[f_\theta(x, y^+)/\tau] - \log \mathbb{E}_{Q_0}[e^{f_\theta(x,y)/\tau}] \right] = -\mathbb{E}_{P_X} \mathbb{E}_{P_0} \left[ \log \frac{e^{f_\theta(x,y^+)/\tau}}{\mathbb{E}_{Q_0}[e^{f_\theta(x,y)/\tau}]} \right].$$

(1)

### 3.1 Motivation

In practical applications of contrastive learning (CL), negative samples $(y)$ are typically drawn uniformly from the training data, which might have similar semantics (for instance, labels). This introduces a potential issue of sampling bias, as posited by Chuang et al. [13], since similar samples are being forcefully separated. Recent research [15; 13] observed that compared to the employment of ideal negative sampling that selects instances with distinctly dissimilar semantics, the existence of sampling bias could result in a significant reduction in performance.

Table 1: **InfoNCE has the ability to mitigate sampling bias.** We compare the performance of various CL methods with/without fine-tuning $\tau$ (marked with $\tau^*/\tau_0$). We also report relative improvements compared with SimCLR($\tau^*$).

| Model | CIFAR10 | | STL10 | |
|---|---|---|---|---|
| | Top-1 | $\tau$ | Top-1 | $\tau$ |
| SimCLR($\tau_0$) | 91.10 | 0.5 | 81.05 | 0.5 |
| SimCLR($\tau^*$) | 92.19 | 0.3 | 87.91 | 0.2 |
| DCL($\tau_0$) | 92.00 (-0.2%) | 0.5 | 84.26 (-4.2%) | 0.5 |
| DCL($\tau^*$) | 92.09 (-0.1%) | 0.3 | 88.20 (+1.0%) | 0.2 |
| HCL($\tau_0$) | 92.12 (-0.0%) | 0.5 | 87.44 (-0.5%) | 0.5 |
| HCL($\tau^*$) | 92.10 (-0.0%) | 0.3 | 87.46 (-0.5%) | 0.2 |

In this study, we observe an intriguing phenomenon where CL inherently demonstrates resilience to sampling bias. We empirically evaluate CL on two benchmark datasets, CIFAR10 and STL10, with the findings detailed in Table 1. Notably, we make the following observations: 1) By fine-tuning the temperature $\tau$, basic SimCLR demonstrates significant improvement, achieving performance levels comparable to methods specifically designed to address sampling bias (*i.e.,* SimCLR($\tau^*$) versus SimCLR($\tau_0$), DCL and HCL); 2) With an appropriately selected $\tau$, the relative improvements realized by DCL and HCL are marginal. These insights lead us to pose two compelling questions: 1) *Why does CL exhibit tolerance to sampling bias?* and 2) *What role does $\tau$ play, and why is it so important?*

### 3.2 Understanding CL from DRO

In this subsection, we first introduce DRO and then analyze CL from the perspective of DRO.

---

[3] Note that our formula adheres to the commonly used InfoNCE expression (e.g., the SimCLR setting [4]), in which the denominator does not include $\mathbb{E}_{P_0}[e^{f_\theta(x,y^+)/\tau}]$. This is also used in Yeh et al. [45].

## A. Distributionally Robust Optimization (DRO).

The success of machine learning heavily relies on the premise that the test and training distributions are identical (*a.k.a. iid* assumption). How to handle different data distributions (*a.k.a.* distribution shift) poses a great challenge to machine learning systems. Fortunately, Distributionally Robust Optimization (DRO) provides a potential solution to mitigate this problem. DRO aims to minimize the worst-case expected loss over a set of potential distributions $Q$, which surround the training distribution $Q_0$ and are constrained by a distance metric $\mathcal{D}\phi$ within a specified robust radius $\eta$. Formally, DRO can be written as:

$$\mathcal{L}_{\text{DRO}} = \max_Q \mathbb{E}_Q[\mathcal{L}(x;\theta)] \qquad s.t. \ D_\phi(Q||Q_0) \leq \eta, \tag{2}$$

where $D_\phi(Q||Q_0)$ measures the $\phi$-divergence between two distributions, and $\mathcal{L}(x;\theta)$ denotes the training loss on input $x$. Intuitively, models incorporating DRO enjoy stronger robustness due to the presence of $Q$ that acts as an "adversary", optimizing the model under a distribution set with adversarial perturbations instead of a single training distribution.

## B. Analyzing CL from DRO.
We first introduce the CL-DRO objective:

**Definition 3.1** (CL-DRO). Let $P_0$ and $Q_0$ be the distributions of positive and negative samples respectively. Let $\eta$ be the robust radius. CL-DRO objective is defined as:

$$\mathcal{L}_{\text{CL-DRO}}^\phi = -\mathbb{E}_{P_X}\left[\mathbb{E}_{P_0}[f_\theta(x,y^+)] - \max_Q \mathbb{E}_Q[f_\theta(x,y)]\right] \qquad s.t. \ D_\phi(Q||Q_0) \leq \eta. \tag{3}$$

The objective CL-DRO can be understood as the DRO enhancement of the basic objective $L_{\text{basic}} = -\mathbb{E}_{P_X}\left[\mathbb{E}_{P_0}[f_\theta(x,y^+)] - \mathbb{E}_{Q_0}[f_\theta(x,y)]\right]$, which aims to increase the embedding similarity between the positive instances and decreases that of the negative ones. CL-DRO imporves $L_{\text{basic}}$ by incorporating DRO on the negative side, where $f_\theta(x,y)$ is optimized a range of potential distributions. Consequently, CL-DRO equips the model with resilience to distributional shifts of negative samples.

**Theorem 3.2.** *By choosing KL divergence $D_{KL}(Q||Q_0) = \int Q \log \frac{Q}{Q_0} dx$, optimizing CL-DRO (cf. Equation* (3)*) is equivalent to optimizing CL (InfoNCE,cf. Equation* (1)*):*

$$\mathcal{L}_{CL\text{-}DRO}^{KL} = -\mathbb{E}_{P_X}\left[\mathbb{E}_{P_0}[f_\theta(x,y^+)] - \min_{\alpha \geq 0, \beta} \max_{Q \in \mathbb{Q}}\{\mathbb{E}_Q[f_\theta(x,y)] - \alpha[D_{KL}(Q||Q_0) - \eta] + \beta(\mathbb{E}_{Q_0}[\frac{Q}{Q_0}] - 1)\}\right]$$

$$= -\mathbb{E}_{P_X}\mathbb{E}_{P_0}\left[\alpha^*(\eta) \log \frac{e^{f_\theta(x,y^+)/\alpha^*(\eta)}}{\mathbb{E}_{Q_0}[e^{f_\theta(x,y)/\alpha^*(\eta)}]}\right] + Constant$$

$$= \alpha^*(\eta)\mathcal{L}_{InfoNCE} + Constant, \tag{4}$$

*where $\alpha, \beta$ represent the Lagrange multipliers, and $\alpha^*(\eta)$ signifies the optimal value of $\alpha$ that minimizes the Equation* (4)*, serving as the temperature $\tau$ in CL.*

The proof is presented in Appendix A.1. Theorem 3.2 admits the merit of CL as it is equivalent to performing DRO on the negative samples. The DRO enables CL to perform well across various potential distributions and thus equips it with the capacity to alleviate sampling bias. We further establish the connection between InfoNCE loss with the unbiased loss that employs ideal negative distribution $L_{\text{unbiased}} = -\mathbb{E}_{P_X}\left[\mathbb{E}_{P_0}[f_\theta(x,y^+)] - \mathbb{E}_{Q^{\text{ideal}}}[f_\theta(x,y)]\right]$.

**Theorem 3.3.** *[Generalization Bound] Let $\widehat{\mathcal{L}}_{InfoNCE}$ be an estimation of InfoNCE with $N$ negative samples. Given any finite hypothesis space $\mathbb{F}$ of models, suppose $f_\theta \in [M_1, M_2]$ and the ideal negative sampling distribution $Q^{ideal}$ satisfies $D_{KL}(Q^{ideal}||Q_0) \leq \eta$, we have that with probability at least $1 - \rho$:*

$$\mathcal{L}_{unbiased} \leq \tau \widehat{\mathcal{L}}_{InfoNCE} + \mathcal{B}(\rho, N, \tau), \tag{5}$$

*where $\mathcal{B}(\rho, N, \tau) = \frac{M_2 \exp((M_2-M_1)/\tau)}{N-1+\exp((M_2-M_1)/\tau)} \sqrt{\frac{N}{2} \ln(\frac{2|\mathbb{F}|}{\rho})}$.*

As we mainly focus on model robustness on negative distribution, here we disregard the constant term present in Equation (4), and omit the error from the positive instances. The proof is presented in

Appendix A.2. Theorem 3.3 exhibits the unbiased loss is upper bound by InfoNCE if a sufficiently large data size is employed (considering $\mathcal{B}(\rho, N, \tau) \to 0$ as $N \to 0$). This bound illustrates how $\tau$ impacts the model performance, which will be detailed in the next subsection.

## 3.3 Understanding $\tau$ from DRO

Building on the idea of DRO, we uncover the role of $\tau$ from the following perspectives:

**A. Adjusting robust radius.** The temperature $\tau$ and the robust radius $\eta$ conform to:

**Corollary 3.4.** *[The optimal $\alpha$ - Lemma 5 of Faury et al. [46]] The value of the optimal $\alpha$ (i.e., $\tau$) can be approximated as follow:*

$$\tau \approx \sqrt{\mathbb{V}_{Q_0}[f_\theta(x,y)]/2\eta}, \tag{6}$$

*where $\mathbb{V}_{Q_0}[f_\theta(x,y)]$ denotes the variance of $f_\theta(x,y)$ under the distribution $Q_0$.*

This corollary suggests that the temperature parameter $\tau$ is a function of the robust radius $\eta$ and the variance of $f_\theta$. In practical applications, tuning $\tau$ in CL is essentially equivalent to adjusting the robustness radius $\eta$.

The aforementioned insight enhances our understanding of the impact of temperature on CL performance. An excessively large value of $\tau$ implies a small robustness radius $\eta$, potentially violating the condition $D_{KL}(Q^{\text{ideal}}||Q_0) \leq \eta$ as stipulated in Theorem 3.2. Intuitively, a large $\tau$ delineates a restricted distribution set in DRO that may fail to encompass the ideal negative distribution, leading to diminished robustness. Conversely, as $\tau$ decreases, the condition $D_{KL}(Q^{\text{ideal}}||Q_0) \leq \eta$ may be satisfied, but it expands the term $\mathcal{B}(\rho, N, \tau)$ and loosens the generalization bound. These two factors establish a trade-off in the selection of $\tau$, which we will empirically validate in Subsection 3.4.

**B. Controlling variance of negative samples.**

**Theorem 3.5.** *Given any $\phi$-divergence, the corresponding CL-DRO objective could be approximated as a mean-variance objective:*

$$\mathcal{L}_{CL\text{-}DRO}^\phi \approx -\mathbb{E}_{P_X}\left[\mathbb{E}_{P_0}[f_\theta(x,y^+)] - \left(\mathbb{E}_{Q_0}[f_\theta(x,y)] + \frac{1}{2\tau}\frac{1}{\phi^{(2)}(1)} \cdot \mathbb{V}_{Q_0}[f_\theta(x,y)]\right)\right], \tag{7}$$

*where $\phi^{(2)}(1)$ denotes the the second derivative value of $\phi(\cdot)$ at point 1, and $\mathbb{V}_{Q_0}[f_\theta]$ denotes variance of $f$ under the distribution $Q_0$.*

*Specially, if we consider KL divergence, the approximation transforms:*

$$\mathcal{L}_{CL\text{-}DRO}^{KL} \approx -\mathbb{E}_{P_X}\left[\mathbb{E}_{P_0}[f_\theta(x,y^+)] - \left(\mathbb{E}_{Q_0}[f_\theta(x,y)] + \frac{1}{2\tau}\mathbb{V}_{Q_0}[f_\theta(x,y)]\right)\right]. \tag{8}$$

The proof is presented in Appendix A.4. Theorem 3.5 provides a *mean-variance* formulation for CL. Compared to the basic objective $L_{\text{basic}}$, CL introduces an additional variance regularization on negative samples, governed by the parameter $\tau$, which modulates the fluctuations in similarity among negative samples. The efficacy of variance regularizers has been extensively validated in recent research [20; 47; 48]. We consider that variance control might be a reason of the success of CL.

**C. Hard-mining.** Note that the worst-case distribution in CL-DRO can be written as $Q^* = Q_0\frac{\exp[f_\theta(x,y^+)/\tau]}{\mathbb{E}_{Q_0}\exp[f_\theta(x,y)/\tau]}$ (refer to Appendix A.6). This implies that the model in CL is optimized under the negative distribution $Q^*$, where each negative instance is re-weighted by $\frac{\exp[f_\theta(x,y^+)/\tau]}{\mathbb{E}_{Q_0}\exp[f_\theta(x,y)/\tau]}$. This outcome vividly exhibits the hard-mining attribute of CL, with $\tau$ governing the degree of skewness of the weight distribution. A smaller $\tau$ suggests a more skewed distribution and intensified hard-mining, while a larger $\tau$ signifies a more uniform distribution and attenuated hard-mining. Our research yields similar conclusions to those found in [16; 28], but we arrive at this finding from a novel perspective.

## 3.4 Empirical Analysis

In this section, we conduct the following experiments to verify the above conclusions.

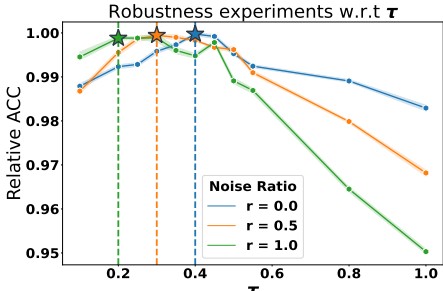

Figure 1: $\tau$ **varies with the level of sampling bias.** We report the relative top1 accuracy of SimCLR *w.r.t.* different $\tau$ across different rate $r$ of false negative samples.

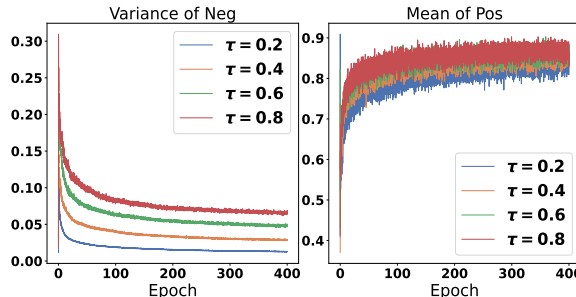

Figure 2: $\tau$ **controls the variance of negative samples score.** We alter the value of $\tau$ and record corresponding variance of negative samples and mean value of positive samples.

$\tau$ **varies with the level of sampling bias.** To study the effect of $\tau$ on model performance, we conduct experiments on the CIFAR10 dataset and manipulate the ratio of false negative samples based on the ground-truth labels. A ratio of 1 implies retaining all false negative instances in the data, while a ratio of 0 implies eliminating all such instances. The detailed experimental setup can refer to the Appendix B. From Figure 1, we make two observations: 1) There is an evident trade-off in the selection of $\tau$. As $\tau$ increases from 0.2 to 1, the model's performance initially improves, then declines, aligning with our theoretical analysis. 2) As the ratio of false negative instances increases ($r$ ranges from 0 to 1), the optimal $\tau$ decreases. This observation is intuitive — a higher ratio of false negative instances implies a larger distribution gap between the employed negative distribution and the ideal, thereby necessitating a larger robust radius $\eta$ to satisfy the condition $D_{KL}(Q^{\text{ideal}}||Q_0) \leq \eta$. As Corollary 3.4, a larger robust radius corresponds to a smaller $\tau$.

$\tau$ **controls the variances of negative samples.** We experiment with an array of $\tau$ values and track the corresponding variance of $f_\theta(x, y)$ for negative samples and the mean value of $f_\theta(x, y^+)$ for positive samples in CIFAR10. As illustrated in Figure 2, we observe a decrease in the variance of negative samples and an increase in the mean value of positive samples as $\tau$ diminishes. Although this observation might be challenging to interpret based on existing studies, it can be easily explained through Equation (8). Note that $\tau$ governs the impact of the variance regularizer. A smaller $\tau$ brings a larger variance penalty, resulting in a reduced variance of negative instances. Simultaneously, as the regularizer's contribution on optimization intensifies, it proportionately diminishes the impact of other loss component on enhancing the similarity of positive instances, thereby leading to a lower mean value of positive instances.

**Mean-Variance objective achieves competitive performance.** We replace the InfoNCE with the mean-variance objective (*cf.* Equation (8)) and evaluate the model performance. The results are shown in Table 2. Remarkably, such a simple loss can can deliver competitive results to InfoNCE, thereby affirming the correctness of the approximation in Theorem 3.5. Nevertheless, it marginally underperforms compared to InfoNCE, as it merely represents a second-order Taylor expansion of InfoNCE. However, it is worth highlighting that the mean-variance objective is more efficient.

Table 2: **Mean-variance objective (MW) achieves comparable performance.** We replace the loss function with variance penalty on negative samples.

| Model | Cifar10 | | Stl10 | |
|---|---|---|---|---|
| | Top-1 | $\tau$ | Top-1 | $\tau$ |
| SimCLR($\tau_0$) | 91.10 | 0.5 | 81.05 | 0.5 |
| SimCLR($\tau^*$) | **92.19** | 0.3 | **87.91** | 0.2 |
| MW | 91.81 | 0.3 | 87.24 | 0.2 |

## 4 Relations among DRO, Constrastive Learning and Mutual Information

Through extensive theoretical analysis, we have far demonstrated that CL is fundamentally a CL-DRO objective that conduct DRO on negative samples. And it is well known that CL is also equivalent to mutual information estimation. A natural question arises: *Is there a connection between CL-DRO and MI, as both are theoretical understanding of CL?*

To make our conclusion more generalized, we would like to transcend the boundary of KL divergence, but exploring a broader family of $\phi$-divergence. First of all, we would like to extend the definition of MI to utilize $\phi$-divergence:

**Definition 4.1** ($\phi$-MI). The $\phi$-divergence-based mutual information is defined as:

$$I_\phi(X;Y) = D_\phi(P(X,Y)||P(X)P(Y)) = \mathbb{E}_{P_X}[D_\phi(P_0||Q_0)]. \tag{9}$$

Comparing to KL-divergence, $\phi$-divergence showcases greater flexibility and may exhibits superior properties. $\phi$-MI, as a potential superior metric for distribution dependence, has been studied by numerous work [49; 50].

**Theorem 4.2.** *For distributions $P$, $Q$ such that $P \ll Q$, let $\mathcal{F}$ be a set of bounded measurable functions. Let CL-DRO draw positive and negative instances from $P$ and $Q$, marked as $\mathcal{L}_{CL\text{-}DRO}^\phi(P,Q)$. Then the CL-DRO objective is the tight variational estimation of $\phi$-divergence. In fact, we have:*

$$D_\phi(P||Q) = \max_{f \in \mathcal{F}} -\mathcal{L}_{CL\text{-}DRO}^\phi(P,Q) = \max_{f \in \mathcal{F}} \mathbb{E}_P[f] - \min_{\lambda \in \mathbb{R}}\{\lambda + \mathbb{E}_Q[\phi^*(f - \lambda)]\}. \tag{10}$$

*Here, the choice of $\phi$ in CL-DRO corresponds to the probability measures in $D_\phi(P||Q)$. And $\phi^*$ denotes the convex conjugate.*

The proof is presented in Appendix A.5. This theorem establishes the connection between CL-DRO and $\phi$-divergence. If we replace $P$ with $P_0$ and $Q$ with $Q_0$, and simply hypothesis that the metric function $f_\theta$ has a sufficiently large capacity, we have the following corollary:

**Corollary 4.3.** $\mathbb{E}_{P_X}[D_\phi(P_0||Q_0)]$ *is a tight variational estimation of $I_\phi(X;Y)$.*

**InfoNCE is a tighter MI estimation.** The prevailing and widely used variational approximation of $\phi$-divergences is the Donsker-Varadhan target ($I_{DV}$), defined as $D_\phi(P||Q) \coloneqq \max_{f \in \mathcal{F}}\{\mathbb{E}_P[f] - \mathbb{E}_Q[\phi^*(f)]\}$. However, it has been observed by Ruderman et al. [51] that this expression is loose when applied to probability measures, as it fails to fully consider the fundamental nature of divergences being defined between probability distributions. To address this limitation, let us further assume, as we do for $\mathbb{R}$, that $\mathbb{E}_P[1] = 1$. Under this common assumption, we arrive at a more refined formulation: $D_\phi(P||Q) \coloneqq \max_{f \in \mathcal{F}}\{\mathbb{E}_P[f] - \min_{\lambda \in \mathbb{R}}\{\lambda + \mathbb{E}_Q[\phi^*(f - \lambda)]\}\}$. Notably, this result, embracing the infimum over $\lambda$ in the spirit of Ruderman et al. [51], appears to be consistent with our proof in Theorem 4.2. Hence, we argue that "InfoNCE is a tighter MI estimation."

Furthermore, this corollary provides a rigorous and succinct methodology for deriving InfoNCE from Variational MI. Specifically, given $\phi(x) = x\log x - x + 1$ for the case of KL-divergence, its conjugate $\phi^*(x) = e^x - 1$ can be computed. Subsequently, by resolving the simple convex problem $\min_{\lambda \in \mathbb{R}}\{\lambda + \tau\mathbb{E}_Q[\phi^*(\frac{f(x,y)-\lambda}{\tau})]\}$, we derive the optimal choice of $\lambda$, denoted as $\lambda^* = \tau\log\mathbb{E}_{Q_0}[e^{f(x,y)/\tau}]$. Inserting $\lambda^*$ into Equation (10), $D_\phi(P_0||Q_0)$ simplifies to $\max_f\{\mathbb{E}_{P_0}[f(x,y^+)] - \tau\log\mathbb{E}_{Q_0}[e^{f(x,y)/\tau}]\}$, which is the minus of the InfoNCE objective.

**DRO bridges the gap between MI and InfoNCE.** Although previous works such as MINE[24] and CPC[3] have demonstrated that InfoNCE is a lower bound for MI, they still have significant limitations [41]. For instance, MINE uses a critic in Donsker-Varadhan target ($\mathcal{I}_{DV}$) to derive a bound that is neither an upper nor lower bound on MI, while CPC relies on unnecessary approximations in its proof, resulting in some redundant approximations. In contrast, Theorem 4.2 presents a rigorous and practical method for accurately estimating the tight lower bound of MI.

**DRO provides general MI estimation.** Although a strict variational bound for MI has been proposed in Poole et al. [41], their discussion is limited to the choice of KL-divergence. Theorem 4.2 is suitable for estimating various $\phi$-MI. For example, if we consider the case of $\chi^2$ divergence, given by $\phi(x) = \frac{1}{2\sqrt{2}}(x-1)^2$, we could obtain convex conjugate $\phi^*(x) = x + x^2$. The variational representation becomes $I_{\chi^2}(X;Y) = D_{\chi^2}(P_0 \| Q_0) = \max_f \{\mathbb{E}_{P_0}[f(x,y^+)] - \mathbb{E}_{Q_0}[f(x,y)] - \mathbb{V}_{Q_0}[f(x,y)]\}$, where $\mathbb{V}_{Q_0}[f(x,y)]$ represents the variance of $f$ on distribution $Q$. Our theoretical framework offers the opportunity to estimate flexible $\phi$-MI, which can be adapted to suit specific scenarios.

## 5 Method

### 5.1 Shortcomings of InfoNCE

Based on the understanding of CL from DRO perspective, some weaknesses of InfoNCE is revealed:

- **Too conservative.** Distributionally Robust Optimization (DRO) focuses on the worst-case distribution, often leading to an overemphasis on the hardest negative samples by assigning them disproportionately large weights. More precisely, the weights are proportional to $\exp[f_\theta(x, y)/\tau]$, causing samples with the highest similarity to receive significantly larger weights. However, in practice, the most informative negative samples are often found in the "head" region (e.g., top-20% region with high similarity) rather than exclusively in the top-1% highest region [52]. Overemphasizing these top instances by assigning excessive weights is typically suboptimal.

- **Sensitive to outliers.** Existing literature [25] reveals that DRO is markedly sensitive to outliers, exhibiting significant fluctuations in training performance. This is because outliers, once they appear in regions of high similarity, can be assigned exceptionally high weights. Consequently, this phenomenon unavoidably influences the rate of model convergence during the training phase.

### 5.2 ADNCE

Our goal is to refine the worst-case distribution, aiming to assign more reasonable weights to negative instances. The aspiration is to create a flexible distribution that can be adjusted to concentrate on the informative region. This is accomplished by incorporating Gaussian-like weights, as defined below:

$$w(f_\theta(x, y), \mu, \sigma) \propto \frac{1}{\sigma\sqrt{2\pi}} \exp[-\frac{1}{2}(\frac{f_\theta(x, y) - \mu}{\sigma})^2], \quad (11)$$

where $\mu$ and $\sigma$ are two hyper-parameter we could control. As is illustrated in Figure 3, $\mu$ controls the central region of weight allocation, whereby samples closer to $\mu$ have larger weights, while $\sigma$ controls the height of the weight allocation in the central region. Intuitively, a smaller $\sigma$ results in a more pronounced weight discrepancy among the samples.

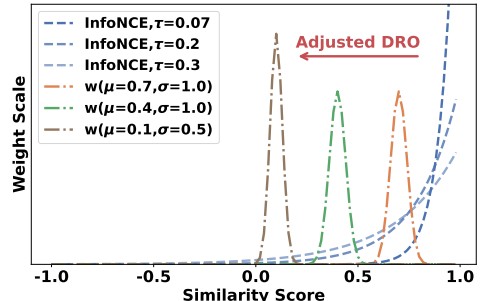

Figure 3: We visualize the training weight of negative samples *w.r.t.* similarity score. InfoNCE (in BlUE) over-emphasize hard negative samples, while ADNCE utilizes the weight $w$ (in ORANGE, GREEN, BROWN) to adjust the distribution.

Inserting the weights into InfoNCE formula, we obtain the novel **AD**justed Info**NCE** (ADNCE):

$$\mathcal{L}_{\text{ADNCE}} = -\mathbb{E}_P[f_\theta(x, y^+)/\tau] + \log \mathbb{E}_{Q_0}[w(f_\theta(x, y), \mu, \sigma)e^{f_\theta(x,y)/\tau}/Z_{\mu,\sigma}], \quad (12)$$

where $Z_{\mu,\sigma}$ denotes the partition function of $w(f(x, y), \mu, \sigma)$, *i.e.*, $Z_{\mu,\sigma} = \mathbb{E}_{Q_0}[w(f_\theta(x, y), \mu, \sigma)]$.

It is noteworthy that our proposed approach only involves two modified lines of code and introduce no additional computational overhead, as compared to InfoNCE [4]. Pytorch-style pseudocode for ADNCE is given in Appendix.

## 6 Experiments

We empirically evaluate our adjusted InfoNCE method — ADNCE, and apply it as a modification to respective classical CL models on image, graph, and sentence data. For all experiments, $\mu$ is treated as a hyper-parameter to adjust the weight allocation, where $\sigma$ is set to 1 by default. For detailed experimental settings, please refer to Appendix.

### 6.1 Image Contrastive Learning

We begin by testing ADNCE on vision tasks using CIFAR10, CIFAR100, and STL10 datasets. We choose SimCLR [4] as the baseline model (termed as InfoNCE($\tau_0$)). We also include two variants

Table 3: Performance comparisons on multiple loss formulations (ResNet50 backbone, batchsize 256). Top-1 accuracy with linear evaluation protocol. $\tau_0$ means $\tau = 0.5$, $\tau^*$ represents grid searching on $\tau$ and $\alpha$-CL-direct is a similar work [16] on CL theory. Bold is highest performance. Each setting is repeated 5 times with different random seeds.

| Model | CIFAR10 | | | | STL10 | | | | CIFAR100 | | | |
|---|---|---|---|---|---|---|---|---|---|---|---|---|
| | 100 | 200 | 300 | 400 | 100 | 200 | 300 | 400 | 100 | 200 | 300 | 400 |
| InfoNCE ($\tau_0$) | 85.70 | 89.21 | 90.33 | 91.01 | 75.95 | 78.47 | 80.39 | 81.67 | 59.10 | 63.96 | 66.03 | 66.53 |
| InfoNCE ($\tau^*$) | 86.54 | 89.82 | 91.18 | 91.64 | 81.20 | 84.77 | 86.27 | 87.69 | 62.32 | 66.85 | 68.31 | 69.03 |
| $\alpha$-CL-direct | 87.65 | 90.11 | 90.88 | 91.24 | 80.91 | 84.71 | 87.01 | 87.96 | 62.75 | 66.27 | 67.35 | 68.54 |
| ADNCE | **87.67** | **90.65** | **91.42** | **91.88** | **81.77** | **85.10** | **87.01** | **88.00** | **62.79** | **66.89** | **68.65** | **69.35** |

Table 4: Sentence embedding performance on STS tasks (Spearman's correlation, "all" setting). **Bold** indicates the best performance while underline indicates the second best on each dataset. ♣: results from SimCSE [9]; all other results are reproduced or reevaluated by ourselves.

| Model | STS12 | STS13 | STS14 | STS15 | STS16 | STS-B | SICK-R | Avg. |
|---|---|---|---|---|---|---|---|---|
| GloVe embeddings (avg.)♣ | 55.14 | 70.66 | 59.73 | 68.25 | 63.66 | 58.02 | 53.76 | 61.32 |
| BERT$_{base}$-flow♣ | 58.40 | 67.10 | 60.85 | 75.16 | 71.22 | 68.66 | 64.47 | 66.55 |
| BERT$_{base}$-whitening♣ | 57.83 | 66.90 | 60.90 | 75.08 | 71.31 | 68.24 | 63.73 | 66.28 |
| CT-BERT$_{base}$♣ | 61.63 | 76.80 | 68.47 | 77.50 | 76.48 | 74.31 | 69.19 | 72.05 |
| SimCSE-BERT$_{base}$($\tau_0$) | 68.40 | **82.41** | 74.38 | 80.91 | 78.56 | 76.85 | **72.23** | 76.25 |
| SimCSE-BERT$_{base}$($\tau^*$) | 71.37 | 81.18 | 74.41 | **82.51** | 79.24 | 78.26 | 70.65 | 76.81 |
| ADNCE-BERT$_{base}$ | **71.38** | 81.58 | **74.43** | 82.37 | **79.31** | **78.45** | 71.69 | **77.03** |
| SimCSE-RoBERTa$_{base}$($\tau_0$) | **70.16** | 81.77 | 73.24 | 81.36 | 80.65 | 80.22 | 68.56 | 76.57 |
| SimCSE-RoBERTa$_{base}$($\tau^*$) | 68.20 | **81.95** | 73.63 | 81.83 | 81.55 | 80.96 | 69.56 | 76.81 |
| ADNCE-RoBERTa$_{base}$ | 69.22 | 81.86 | **73.75** | **82.88** | **81.88** | **81.13** | **69.57** | **77.10** |

of vanilla InfoNCE for comprehensive comparisons. One approach is applying a grid search for $\tau$ (termed as InfoNCE($\tau^*$)), which serves to confirm its importance as discussed in Section 3.4. The other is from Tian el al. [16], which proposes a novel approach for setting for the value of $\tau$ (termed as $\alpha$-CL-direct).

As shown in Table 3, a grid search for $\tau$ has a crucial impact on the model performance. This impact is significant throughout the entire training process, from the early stage (100 epochs) to the later stage (400 epochs). Additionally, ADNCE exhibits sustained improvement and notably enhances performance in the early stages of training. In contrast, while $\alpha$-CL-direct introduces a novel approach for setting the value of $\tau$, its essence remains weighted towards the most difficult negative samples, hence yielding similar performance compared to fine-tuning $\tau$ using grid search. In Figure 4, we plot the training curve to further illustrate the stable superiority of ADNCE.

Table 5: **Self-supervised representation learning on TUDataset**: The baseline results are excerpted from the published papers.

| Methods | RDT-B | NCI1 | PROTEINS | DD |
|---|---|---|---|---|
| node2vec | - | 54.9±1.6 | 57.5±3.6 | - |
| sub2vec | 71.5±0.4 | 52.8±1.5 | 53.0±5.6 | - |
| graph2vec | 75.8±1.0 | 73.2±1.8 | 73.3±2.1 | - |
| InfoGraph | 82.5±1.4 | 76.2±1.1 | 74.4±0.3 | 72.9±1.8 |
| JOAO | 85.3±1.4 | 78.1±0.5 | 74.6±0.4 | 77.3±0.5 |
| JOAOv2 | 86.4±1.5 | 78.4±0.5 | 74.1±1.1 | 77.4±1.2 |
| RINCE | 90.9±0.6 | 78.6±0.4 | 74.7±0.8 | 78.7±0.4 |
| GraphCL ($\tau_0$) | 89.5±0.8 | 77.9±0.4 | 74.4±0.5 | 78.6±0.4 |
| GraphCL ($\tau^*$) | 90.7±0.6 | 79.2±0.3 | 74.7±0.6 | 78.5±1.0 |
| ADNCE | **91.4±0.3** | **79.3±0.7** | **75.1±0.6** | **79.2±0.6** |

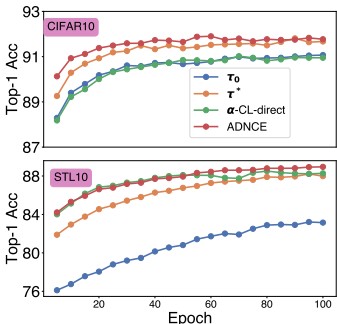

Figure 4: Learning curve for Top-1 accuracy by linear evaluation on CIFAR10 and STL10.

### 6.2 Sentence Contrastive Learning

We follow the experimental setup in GraphCL [12] and conduct evaluations on 7 semantic textual similarity (STS) tasks in an unsupervised setting. Specifically, we evaluate the model on STS 2012-2016, STS-benchmark, and SICK-Relatedness. To ensure fair comparisons, we use pre-trained checkpoints of BERT and RoBERTa, and randomly sample sentences from the English Wikipedia.

As observed in Table 4, ADNCE consistently outperforms InfoNCE, achieving an average Spearman's correlation of 77%. The ease of replacing InfoNCE with ADNCE and the resulting substantial performance improvements observed in BERT and RoBERTa serve as testaments to the efficacy and broad applicability of ADNCE. Furthermore, the improvements of $\tau^*$ over $\tau_0$ emphasize the significance of selecting a proper robustness radius.

### 6.3 Graph Contrastive Learning

To study the modality-agnostic property of ADNCE beyond images and sentences, we conduct an evaluation of its performance on TUDataset [53]. We use a classical graph CL model GraphCL [12] as the baseline. To ensure a fair comparison, we adhered to the same protocol used in GraphCL [12], employing techniques such as node dropout, edge perturbation, attribute masking, and subgraph sampling for data augmentation. We replace the vanilla InfoNCE with two different variants, namely GraphCL($\tau^*$) obtained through grid search, and our proposed ADNCE.

Table 5 demonstrates that ADNCE outperforms all baselines with a significant margin on four datasets, especially when compared to three state-of-the-art InfoNCE-based contrastive methods, GraphCL, JOAO, and JOAOv2, thereby setting new records on all four datasets. The improvements observed in GraphCL($\tau^*$) relative to GraphCL($\tau_0$) align with our understanding of DRO.

## 7 Conclusion and Limitations

We provide a novel perspective on contrastive learning (CL) via the lens of Distributionally Robust Optimization (DRO), and reveal several key insights about the tolerance to sampling bias, the role of $\tau$, and the theoretical connection between DRO and MI. Both theoretical analyses and empirical experiments confirm the above findings. Furthermore, we point out the potential shortcomings of CL from the perspective of DRO, such as over-conservatism and sensitivity to outliers. To address these issues, we propose a novel CL loss — ADNCE, and validate its effectiveness in various domains.

The limitations of this work mainly stem from two aspects: 1) Our DRO framework only provides a theoretical explanation for InfoNCE-based methods, leaving a crucial gap in the success of CL without negative samples [4; 26]. 2) ADNCE requires weight allocation to be adjusted through parameters and cannot adaptively learn the best reweighting scheme.

## Acknowledgments and Disclosure of Funding

This work is supported by the National Key Research and Development Program of China (2021ZD0111802), the National Natural Science Foundation of China (62372399, 9227010114, 62302321), the Starry Night Science Fund of Zhejiang University Shanghai Institute for Advanced Study (SN-ZJU-SIAS-001), and supported by the University Synergy Innovation Program of Anhui Province (GXXT-2022-040).

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

# A    Appendix of Proofs

## A.1    Proof of Theorem 3.2

**Theorem 3.2.** *By choosing KL divergence $D_{KL}(Q||Q_0) = \int Q \log \frac{Q}{Q_0} dx$, optimizing CL-DRO (cf. Equation (3)) is equivalent to optimizing CL (InfoNCE,cf. Equation (1)):*

$$
\begin{aligned}
\mathcal{L}_{\text{CL-DRO}}^{KL} &= -\mathbb{E}_{P_X}\left[\mathbb{E}_{P_0}[f_\theta(x,y^+)] - \min_{\alpha \geq 0, \beta} \max_{Q \in \mathbb{Q}}\{\mathbb{E}_Q[f_\theta(x,y)] - \alpha[D_{KL}(Q||Q_0) - \eta] + \beta(\mathbb{E}_{Q_0}[\frac{Q}{Q_0}] - 1)\}\right] \\
&= -\mathbb{E}_{P_X}\mathbb{E}_{P_0}\left[\alpha^*(\eta) \log \frac{e^{f_\theta(x,y^+)/\alpha^*(\eta)}}{\mathbb{E}_{Q_0}[e^{f_\theta(x,y)/\alpha^*(\eta)}]}\right] + Constant \\
&= \alpha^*(\eta)\mathcal{L}_{\text{InfoNCE}} + Constant,
\end{aligned}
$$

(4)

*where $\alpha, \beta$ represent the Lagrange multipliers, and $\alpha^*(\eta)$ signifies the optimal value of $\alpha$ that minimizes the Equation (4), serving as the temperature $\tau$ in CL.*

*Proof.* **In all the following appendix, for the sake of conserving space and clarity of expression, we have opted to simplify $f_\theta(x,y)$ as $f_\theta$ and omit $\mathbb{E}_{P_X}$.**

To complete the proof, we start with giving some important notations and theorem.

**Definition A.1** ($\phi$-divergence [54]). For any convex funtion $\phi$ with $\phi(1) = 0$, the $\phi$-divergence between $Q$ and $Q_0$ is:

$$D_\phi(Q||Q_0) \coloneqq \mathbb{E}_{Q_0}[\phi(dQ/dQ_0)]$$

(13)

where $D_\phi(Q||Q_0) = \infty$ if $Q$ is not absolutely continuous with respect to $Q_0$. Specially, when $\phi(x) = x \log x - x + 1$, $\phi$-divergence degenerates to the well-known KL divergence.

**Definition A.2** (Convex conjugate [55]). We consider a pair $(A, B)$ of topological vector spaces and a bilinear form $\langle \cdot, \cdot \rangle \to \mathbb{R}$ such that $(A, B, \langle \cdot, \cdot \rangle)$ form a dual pair. For a convex function $f : \mathbb{R} \to \mathbb{R}$, $dom f \coloneqq \{x \in \mathbb{R} : f(x) < \infty\}$ is the effective domain of $f$. The convex conjugate, also known as the Legendre-Fenchel transform, of $f : A \to \mathbb{R}$ is the function $f^* : B \to \mathbb{R}$ defined as

$$f^*(b) = \sup_a \{ab - f(a)\}, \quad b \in B$$

(14)

**Theorem A.3** (Interchange of minimization and integration [56]). *Let $(\Omega, \mathcal{F})$ be a measurable space equipped with $\sigma$-algebra $\mathcal{F}$, $L^p(\Omega, \mathcal{F}, P)$ be the linear space of measurable real valued functions $f : \Omega \to \mathbb{R}$ with $||f||_p < \infty$, and let $\mathcal{X} \coloneqq L^p(\Omega, \mathcal{F}, P)$, $p \in [1, +\infty]$. Let $g : \mathbb{R} \times \Omega \to \mathbb{R}$ be a normal integrand, and define on $\mathcal{X}$. Then,*

$$\min_{x \in \mathcal{X}} \int_\Omega g(x(\omega), \omega) \, dP(\omega) = \int_\Omega \min_{s \in \mathbb{R}} g(s, \omega) \, dP(\omega)$$

(15)

To ease the derivation, we denote the likelihood raito $L(x,y) = Q(x,y)/Q_0(x,y)$. Note that the $\phi$-divergence between $Q$ and $Q_0$ is constrained, and thus $L(.)$ is well-defined. For brevity, we usually short $L(x,y)$ as $L$. And in terms of Definition A.1 of $\phi$-divergence, the expression of CL-DRO becomes:

$$\mathcal{L}_{\text{CL-DRO}}^\phi = -\mathbb{E}_{P_0}[f_\theta] + \max_L \mathbb{E}_{Q_0}[f_\theta L] \qquad s.t. \ \mathbb{E}_{Q_0}[\phi(L)] \leq \eta$$

(16)

Note that $\mathbb{E}_{Q_0}[f_\theta L]$ and $\mathbb{E}_{Q_0}[\phi(L)]$ are both convex in $L$. We use the Lagrangian function solver:

$$
\begin{aligned}
\mathcal{L}_{\text{CL-DRO}}^\phi &= -\mathbb{E}_{P_0}[f_\theta] + \min_{\alpha \geq 0, \beta} \max_L \{\mathbb{E}_{Q_0}[f_\theta L] - \alpha[\mathbb{E}_{Q_0}[\phi(L)] - \eta] + \beta(\mathbb{E}_{Q_0}[L] - 1)\} \\
&= -\mathbb{E}_{P_0}[f_\theta] + \min_{\alpha \geq 0, \beta} \left\{\alpha\eta - \beta + \alpha \max_L\{\mathbb{E}_{Q_0}[\frac{f_\theta + \beta}{\alpha}L - \phi(L)]\}\right\} \\
&= -\mathbb{E}_{P_0}[f_\theta] + \min_{\alpha \geq 0, \beta} \left\{\alpha\eta - \beta + \alpha\mathbb{E}_{Q_0}[\max_L\{\frac{f_\theta + \beta}{\alpha}L - \phi(L)\}]\right\} \\
&= -\mathbb{E}_{P_0}[f_\theta] + \min_{\alpha \geq 0, \beta} \left\{\alpha\eta - \beta + \alpha\mathbb{E}_{Q_0}[\phi^*(\frac{f_\theta + \beta}{\alpha})]\right\}
\end{aligned}
$$

(17)

The first equality holds due to the strong duality [57]. The second equality is a re-arrangement for optmizing $L$. The third equation follows by the Theorem A.3. The last equality is established based

on the definition of convex conjugate A.2. When we choose KL-divergence, we have $\phi_{KL}(x) = x \log x - x + 1$. It can be deduced that $\phi_{KL}^*(x) = e^x - 1$. Then, we have:

$$
\begin{aligned}
\mathcal{L}_{\text{CL-DRO}}^{KL} &= -\mathbb{E}_{P_0}[f_\theta] + \min_{\alpha \geq 0, \beta} \left\{ \alpha \eta - \beta + \alpha \mathbb{E}_{Q_0}[\phi^*(\frac{f_\theta + \beta}{\alpha})] \right\} \\
&= -\mathbb{E}_{P_0}[f_\theta] + \min_{\alpha \geq 0, \beta} \left\{ \alpha \eta - \beta + \alpha \mathbb{E}_{Q_0}[e^{\frac{f_\theta + \beta}{\alpha}} - 1] \right\} \\
&= -\mathbb{E}_{P_0}[f_\theta] + \min_{\alpha \geq 0} \left\{ \alpha \eta + \alpha \log \mathbb{E}_{Q_0}[e^{\frac{f_\theta}{\alpha}}] \right\} \\
&= -\mathbb{E}_{P_0}[f_\theta] + \min_{\alpha \geq 0} \left\{ \alpha \eta + \alpha \log \mathbb{E}_{Q_0}[e^{\frac{f_\theta}{\alpha}}] \right\} \\
&= -\mathbb{E}_{P_0}\left[ \alpha^* \log \frac{e^{f_\theta / \alpha^*}}{\mathbb{E}_{Q_0}[e^{f_\theta / \alpha^*}]} \right] + \alpha \eta \\
&= \alpha^* \mathcal{L}_{\text{InfoNCE}} + Constant
\end{aligned}
\tag{18}
$$

Here $\alpha^* = \arg \min_{\alpha \geq 0} \left\{ \alpha \eta + \alpha \log \mathbb{E}_{Q_0}[e^{\frac{f_\theta}{\alpha}}] \right\}$. $\qquad \square$

## A.2 Proof of Theorem 3.3

**Theorem 3.3.** *[Generalization Bound] Let $\widehat{\mathcal{L}}_{InfoNCE}$ be an estimation of InfoNCE with $N$ negative samples. Given any finite hypothesis space $\mathbb{F}$ of models, suppose $f_\theta \in [M_1, M_2]$ and the ideal negative sampling distribution $Q^{ideal}$ satisfies $D_{KL}(Q^{ideal}||Q_0) \leq \eta$, we have that with probability at least $1 - \rho$:*

$$
\mathcal{L}_{unbiased} \leq \tau \widehat{\mathcal{L}}_{InfoNCE} + \mathcal{B}(\rho, N, \tau),
\tag{5}
$$

*where $\mathcal{B}(\rho, N, \tau) = \frac{M_2 \exp((M_2 - M_1)/\tau)}{N - 1 + \exp((M_2 - M_1)/\tau)} \sqrt{\frac{N}{2} \ln(\frac{2|\mathbb{F}|}{\rho})}$.*

Here we simply disregard the constant term present in Equation (4) as it does not impact optimization, and omit the error from the positive instances.

*Proof.* Before detailing the proof process, we first introduce a pertinent theorem:

**Theorem A.4** (Union Bound). *For any two sets $A, B$ and a distribution $\mathcal{D}$ we have*

$$
\mathcal{D}(A \cup B) \leq \mathcal{D}(A) + \mathcal{D}(B)
\tag{19}
$$

**Theorem A.5** (McDiarmid's inequality [58]). *Let $X_1, \cdots, X_n$ be independent random variables, where $X_i$ has range $\mathcal{X}$. Let $f : \mathcal{X}_1 \times \cdots \times \mathcal{X}_n \to \mathbb{R}$ be any function with the $(c_1, \ldots, c_n)$-bounded difference property: for every $i = 1, \ldots, n$ and every $(x_1, \ldots, x_n), (x'_1, \ldots, x'_n) \in \mathcal{X}_1 \times \cdots \times \mathcal{X}_n$ that differ only in the $i$-th coordinate $(x_j = x'_j$ for all $j \neq i)$, we have $|f(x_1, \ldots, x_n) - f(x'_1, \ldots, x'_n)| \leq c_i$. For any $\epsilon > 0$,*

$$
\mathbb{P}(f(X_1, \cdots, X_n) - \mathbb{E}[f(X_1, \cdots, X_n)] \geq \epsilon) \leq \exp(\frac{-2\epsilon^2}{\sum_{i=1}^N c_i^2})
\tag{20}
$$

Now we delve into the proof. As $Q^{ideal}$ satisfies $D_{KL}(Q||Q_0) \leq \eta$, we can bound $\mathcal{L}_{unbiased}$ with:

$$
\begin{aligned}
\mathcal{L}_{unbiased} &= -\mathbb{E}_{P_0}[f_\theta] + \mathbb{E}_{Q^{ideal}}[f_\theta] \\
&\leq -\mathbb{E}_{P_0}[f_\theta] + \max_{D_{KL}(Q||Q_0) \leq \eta} \mathbb{E}_Q[f_\theta] \\
&= \mathcal{L}_{\text{CL-DRO}}^{KL}
\end{aligned}
\tag{21}
$$

where $Q^{ideal}, Q^*$ denotes the ideal negative distribution and the worst-case distribution in CL-DRO. From the Theorem 3.2, we have the equivalence between InfoNCE and CL-DRO. Thus here we

choose CL-DRO for analyses. Suppose we have $N$ negative samples, and for any pair of samples $(x_i, y_i), (x_j, y_j)$, we have the following bound:

$$|Q^*(x_i, y_i)f_\theta(x_i, y_i) - Q^*(x_j, y_j)f_\theta(x_j, y_j)| \leq \sup_{(x,y)\sim Q_0} |Q^*(x,y)f_\theta(x,y)| \leq \frac{\exp\left(\frac{1}{\tau}\right)}{N-1+\exp\left(\frac{1}{\tau}\right)} \tag{22}$$

where the first inequality holds as $Q^*(x,y)f_\theta(x,y) > 0$. The second inequality holds based on the expression of $Q^* = Q_0 \frac{\exp[f_\theta/\tau]}{E_{Q_0}\exp[f_\theta/\tau]}$ (refer to Appendix A.6). Suppose $f_\theta \in [M_1, M_2]$, the upper bound of $\sup_{(x,y)\sim Q_0} |Q^*(x,y)f_\theta(x,y)|$ arrives if $f_\theta(x,y) = M_2$ for the sample $(x,y)$ and $f_\theta(x,y) = M_1$ for others. We have $\sup_{(x,y)\sim Q_0} |Q^*(x,y)f_\theta(x,y)| \leq \frac{M_2\exp((M_2-M_1)/\tau)}{N-1+\exp((M_2-M_1)/\tau)}$.

By using McDiarmid's inequality in Theorem A.5,for any $\epsilon$, we have:

$$\mathbb{P}[(\mathcal{L}_{\text{CL-DRO}}^{KL} - \tau\widehat{\mathcal{L}}_{\text{InfoNCE}}) \geq \epsilon] \leq \exp\left(\frac{-2\epsilon^2}{N}\left(\frac{N-1+\exp((M_2-M_1)/\tau)}{M_2\exp((M_2-M_1)/\tau)}\right)^2\right) \tag{23}$$

Equation (23) holds for the unique model $f$. For any finite hypothesis space of models $\mathbb{F} = \{f_1, f_2, \cdots, f_{|\mathbb{F}|}\}$, we combine this with Theorem A.4 and have following generalization error bound of the learned model $\hat{f}$:

$$\mathbb{P}[(\mathcal{L}_{\text{CL-DRO}}^{KL} - \tau\widehat{\mathcal{L}}_{\text{InfoNCE}}) \geq \epsilon] \leq \sum_{f\in\mathbb{F}} \exp\left(\frac{-2\epsilon^2}{N}\left(\frac{N-1+\exp((M_2-M_1)/\tau)}{M_2\exp((M_2-M_1)/\tau)}\right)^2\right)$$

$$= |\mathbb{F}|\exp\left(\frac{-2\epsilon^2}{N}\left(\frac{N-1+\exp((M_2-M_1)/\tau)}{M_2\exp((M_2-M_1)/\tau)}\right)^2\right) \tag{24}$$

Let

$$\rho = |\mathbb{F}|\exp\left(\frac{-2\epsilon^2}{N}\left(\frac{N-1+\exp((M_2-M_1)/\tau)}{M_2\exp((M_2-M_1)/\tau)}\right)^2\right) \tag{25}$$

we get:

$$\epsilon = \frac{M_2\exp\left((M_2-M_1)/\tau\right)}{N-1+\exp\left((M_2-M_1)/\tau\right)}\sqrt{\frac{N}{2}\ln(\frac{2|\mathbb{F}|}{\rho})} \tag{26}$$

Thus, for $\forall \rho \in (0,1)$, we conclude that with probability at least $1-\rho$.

$$\mathcal{L}_{\text{unbiased}} \leq \mathcal{L}_{\text{CL-DRO}}^{KL} \leq \tau\widehat{\mathcal{L}}_{\text{InfoNCE}} + \frac{M_2\exp\left((M_2-M_1)/\tau\right)}{N-1+\exp\left((M_2-M_1)/\tau\right)}\sqrt{\frac{N}{2}\ln(\frac{2|\mathbb{F}|}{\rho})} \tag{27}$$

$\square$

## A.3   Proof of Corollary 3.4

**Corollary 3.4.** *[The optimal $\alpha$ - Lemma 5 of Faury et al. [46]] The value of the optimal $\alpha$ (i.e., $\tau$) can be approximated as follow:*

$$\tau \approx \sqrt{\mathbb{V}_{Q_0}[f_\theta(x,y)]/2\eta}, \tag{6}$$

*where $\mathbb{V}_{Q_0}[f_\theta(x,y)]$ denotes the variance of $f_\theta(x,y)$ under the distribution $Q_0$.*

*Proof.* While Corollary 3.4 has already been proven in [46], we present a brief outline of the proof here for the sake of completeness and to ensure that our article is self-contained. To verify the relationship between $\tau$ and $\eta$, we could utilize the approximate expression of InfoNCE (*cf.* Equation (31)) and focus on the first order conditions for $\tau$. In detail, we have:

$$-\mathbb{E}_{P_0}[f_\theta] + \inf_{\alpha\geq 0}\{\mathbb{E}_{Q_0}[f_\theta] + \frac{1}{2\alpha}\frac{1}{\phi^{(2)}(1)}\mathbb{V}_{Q_0}[f_\theta] + \alpha\eta\}$$

To find the optimal value of $\alpha$ (or equivalently, $\tau$), we differentiate the above equation and set it to 0. This yields a fixed-point equation

$$\tau = \sqrt{\frac{\mathbb{V}_{Q_0}[f_\theta]}{2\eta}}$$

The corollary gets proved. $\square$

### A.4 Proof of Theorem 3.5

**Theorem 3.5.** *Given any $\phi$-divergence, the corresponding CL-DRO objective could be approximated as a mean-variance objective:*

$$\mathcal{L}^{\phi}_{CL\text{-}DRO} \approx -\mathbb{E}_{P_X}\left[\mathbb{E}_{P_0}[f_\theta(x,y^+)] - \left(\mathbb{E}_{Q_0}[f_\theta(x,y)] + \frac{1}{2\tau}\frac{1}{\phi^{(2)}(1)} \cdot \mathbb{V}_{Q_0}[f_\theta(x,y)]\right)\right], \quad (7)$$

*where $\phi^{(2)}(1)$ denotes the the second derivative value of $\phi(\cdot)$ at point 1, and $\mathbb{V}_{Q_0}[f_\theta]$ denotes the variance of $f$ under the distribution $Q_0$.*

*Specially, if we consider KL divergence, the approximation transforms:*

$$\mathcal{L}^{KL}_{CL\text{-}DRO} \approx -\mathbb{E}_{P_X}\left[\mathbb{E}_{P_0}[f_\theta(x,y^+)] - \left(\mathbb{E}_{Q_0}[f_\theta(x,y)] + \frac{1}{2\tau}\mathbb{V}_{Q_0}[f_\theta(x,y)]\right)\right]. \quad (8)$$

*Proof.* We start with introducing a useful lemma.

**Lemma A.6** (Lemma A.2 of [47]). *Suppose that $\phi : \mathbb{R} \to \mathbb{R} \bigcup\{+\infty\}$ is a closed, convex function such that $\phi(z) \geq \phi(1) = 0$ for all $z$, is two times continuously differentiable around $z = 1$, and $\phi(1) > 0$, Then*

$$\begin{aligned}
\phi^*(\zeta) &= \max_z\{z\zeta - \phi(z)\} \\
&= \zeta + \frac{1}{2!}\left[\frac{1}{\phi''(1)}\right]\zeta^2 + o(\zeta^2)
\end{aligned} \quad (28)$$

Note that most of the $\phi$-divergences [54] (*e.g.,* KL divergence, Cressie-Read divergence, Burg entropy, J-divergence, $\chi^2$-distance, modified $\chi^2$-distance, and Hellinger distance) satisfy the smoothness conditions. When $n = 2$, $\phi^*[\zeta] \approx \zeta + \frac{1}{2}\left[\frac{1}{\phi^{(2)}(1)}\right]\zeta^2$. Substituting this back to Equation (17) we have:

$$\begin{aligned}
\mathcal{L}^{\phi}_{CL\text{-}DRO} &= -\mathbb{E}_{P_0}[f_\theta] + \min_{\alpha \geq 0, \beta}\left\{\alpha\eta - \beta + \alpha\mathbb{E}_{Q_0}[\phi^*(\frac{f_\theta + \beta}{\alpha})]\right\} \\
&= -\mathbb{E}_{P_0}[f_\theta] + \min_{\alpha \geq 0, \beta}\left\{\alpha\eta - \beta + \alpha\mathbb{E}_{Q_0}[\frac{f_\theta + \beta}{\alpha} + \frac{1}{2}\frac{1}{\phi^{(2)}(1)}(\frac{f_\theta + \beta}{\alpha})^2]\right\} \\
&= -\mathbb{E}_{P_0}[f_\theta] + \min_{\alpha \geq 0, \beta}\left\{\alpha\eta - \beta + \mathbb{E}_{Q_0}[f_\theta + \beta + \frac{1}{2}\frac{1}{\phi^{(2)}(1)\alpha}(f_\theta + \beta)^2]\right\} \\
&= -\mathbb{E}_{P_0}[f_\theta] + \min_{\alpha \geq 0, \beta}\left\{\alpha\eta + \mathbb{E}_{Q_0}[f_\theta + \frac{1}{2}\frac{1}{\phi^{(2)}(1)\alpha}(f_\theta + \beta)^2]\right\}
\end{aligned} \quad (29)$$

If we differentiate it *w.r.t.* $\beta$:

$$\frac{\partial\left\{\alpha\eta + \mathbb{E}_{Q_0}[f_\theta + \frac{1}{2}\frac{1}{\phi^{(2)}(1)\alpha}(f_\theta + \beta)^2]\right\}}{\partial\beta} = 0 \quad (30)$$

we have $\beta^* = -\mathbb{E}_{Q_0}[f_\theta]$, and the objective transforms into:

$$\begin{aligned}
&-\mathbb{E}_{P_0}[f_\theta] + \inf_{\alpha \geq 0, \beta}\left\{\alpha\eta + \mathbb{E}_{Q_0}[f_\theta + \frac{1}{2}\frac{1}{\phi^{(2)}(1)\alpha}(f_\theta + \beta)^2]\right\} \\
&= -\mathbb{E}_{P_0}[f_\theta] + \inf_{\alpha \geq 0}\{\mathbb{E}_{Q_0}[f_\theta + \frac{1}{2\alpha}\frac{1}{\phi^{(2)}(1)}(f_\theta - \mathbb{E}_{Q_0}[f_\theta])^2] + \alpha\eta\} \\
&= -\mathbb{E}_{P_0}[f_\theta] + \inf_{\alpha \geq 0}\{\mathbb{E}_{Q_0}[f_\theta] + \frac{1}{2\alpha}\frac{1}{\phi^{(2)}(1)}\mathbb{V}_{Q_0}[f_\theta] + \alpha\eta\}
\end{aligned} \quad (31)$$

Choosing KL-divergence, we have $\phi^{(2)}(1) = 1$. Substituting $\alpha^*(\tau)$ into Equation (31) and ignoring the constant $\alpha\eta$:

$$-\mathbb{E}_{P_0}[f_\theta] + \mathbb{E}_{Q_0}[f_\theta] + \frac{1}{2\tau}\mathbb{V}_{Q_0}[f_\theta]$$

Then Theorem gets proved. $\qquad\qquad\square$

## A.5 Proof of Theorem 4.2

**Theorem 4.2.** *For distributions $P, Q$ such that $P \ll Q$, let $\mathcal{F}$ be a set of bounded measurable functions. Let CL-DRO draw positive and negative instances from $P$ and $Q$, marked as $\mathcal{L}_{CL\text{-}DRO}^{\phi}(P, Q)$. Then the CL-DRO objective is the tight variational estimation of $\phi$-divergence. In fact, we have:*

$$D_{\phi}(P||Q) = \max_{f \in \mathcal{F}} -\mathcal{L}_{CL\text{-}DRO}^{\phi}(P, Q) = \max_{f \in \mathcal{F}} \mathbb{E}_P[f] - \min_{\lambda \in \mathbb{R}}\{\lambda + \mathbb{E}_Q[\phi^*(f - \lambda)]\}. \tag{10}$$

*Here, the choice of $\phi$ in CL-DRO corresponds to the probability measures in $D_{\phi}(P||Q)$. And $\phi^*$ denotes the convex conjugate.*

*Proof.* Regarding this theorem, our proof primarily relies on the variational representation of $\phi$-divergence and optimized certainty equivalent (OCE) risk. Towards this end, we start to introduce the basic concepts:

**Definition A.7** (OCE [56]). Let $X$ be a random variable and let $u$ be a convex, lower-semicontinuous function satisfies $u(0) = 0, u^*(1) = 0$, then optimized certainty equivalent (OCE) risk $\rho(X)$ is defines as:

$$\rho(X) = \inf_{\lambda \in \mathbb{R}}\{\lambda + \mathbb{E}[u(f - \lambda)]\} \tag{32}$$

OCE is a type of risk measure that is widely used by both practitioners and academics [59; 56]. With duality theory, its various properties have been inspiring in our study of DRO.

**Definition A.8** (Variational formulation).

$$D_{\phi}(P||Q) := \sup_{f \in \mathcal{F}}\{\mathbb{E}_P[f] - \mathbb{E}_Q[\phi^*(f)]\} \tag{33}$$

where the supremum is taken over all bounded real-valued measurable functions $\mathcal{F}$ defined on $\mathcal{X}$.

Note that in order to keep consistent with the definition of CL-DRO, we transform Equation (10) to :

$$D_{\phi}(P||Q) = \max_{f \in \mathcal{F}} -\mathcal{L}_{\text{CL-DRO}}^{\phi}(P, Q) = \max_{f \in \mathcal{F}} \mathbb{E}_P[f] - \min_{\beta \in \mathbb{R}}\{-\beta + \mathbb{E}_Q[\phi^*(f + \beta)]\} \tag{34}$$

Our proof for this theorem primarily relies on utilizing OCE risk as a bridge and can be divided into two distinct steps:

Step 1: $\max_{f \in \mathcal{F}} -\mathcal{L}_{\text{CL-DRO}}^{\phi}(P, Q) = \max_{f \in \mathcal{F}} \mathbb{E}_P[f] - \min_{\beta \in \mathbb{R}}\{-\beta + \mathbb{E}_Q[\phi^*(f + \beta)]\}$.

Step 2: $D_{\phi}(P||Q) = \max_{f \in \mathcal{F}} \mathbb{E}_P[f] - \min_{\beta \in \mathbb{R}}\{-\beta + \mathbb{E}_Q[\phi^*(f + \beta)]\}$

1. **We show that** $\max_{f \in \mathcal{F}} -\mathcal{L}_{\textbf{CL-DRO}}^{\phi}(P, Q) = \max_{f \in \mathcal{F}} \mathbb{E}_P[f] - \min_{\beta \in \mathbb{R}}\{-\beta + \mathbb{E}_Q[\phi^*(f + \beta)]\}$.

$$\begin{aligned}
-\mathcal{L}_{\text{CL-DRO}}^{\phi} &= \mathbb{E}_P[f] - \min_{\alpha \geq 0, \beta} \max_L \{\mathbb{E}_Q[fL] - \alpha[\mathbb{E}_Q[\phi(L)] - \eta] + \beta(\mathbb{E}_Q[L] - 1)\} \\
&= \mathbb{E}_P[f] - \min_{\alpha \geq 0, \beta}\{\alpha\eta - \beta + \alpha\mathbb{E}_Q[\max_L\{\frac{f + \beta}{\alpha}L - \phi(L)\}]\} \\
&= \mathbb{E}_P[f] - \min_{\alpha \geq 0, \beta}\{\alpha\eta - \beta + \alpha\mathbb{E}_Q[\phi^*(\frac{f + \beta}{\alpha})]\} \\
&= \mathbb{E}_P[f] - \min_{\beta}\{\alpha^*\eta - \beta + \alpha^*\mathbb{E}_Q[\phi^*(\frac{f + \beta}{\alpha^*})]\} \\
&= \mathbb{E}_P[f] - \min_{\beta}\{-\beta + \alpha^*\mathbb{E}_Q[\phi^*(\frac{f + \beta}{\alpha^*})] + Constant\}
\end{aligned} \tag{35}$$

When $\alpha^* = 1$, step 1 gets proved.

2. **We show that** $D_\phi(P||Q) = \max_{f \in \mathcal{F}} \mathbb{E}_P[f] - \min_{\beta \in \mathbb{R}}\{-\beta + \mathbb{E}_Q[\phi^*(f + \beta)]\}$.

Firstly, we transform $\mathbb{E}_P[f]$ to $\mathbb{E}_Q[f\frac{dP}{dQ}]$ as:

$$
\begin{aligned}
&\max_{f \in \mathcal{F}} \mathbb{E}_P[f] - \min_\beta \left\{ -\beta + \mathbb{E}_Q[\phi^*(f + \beta)] \right\} \\
=&\max_{f \in \mathcal{F}} \mathbb{E}_Q[f\frac{dP}{dQ}] - \min_\beta \left\{ -\beta + \mathbb{E}_Q[\phi^*(f + \beta)] \right\}
\end{aligned}
\tag{36}
$$

Let $f + \beta = Y$, we have:

$$
\begin{aligned}
&\max_{f \in \mathcal{F}} \mathbb{E}_Q[f\frac{dP}{dQ}] - \min_\beta \left\{ -\beta + \mathbb{E}_Q[\phi^*(f + \beta)] \right\} \\
=&\max_{Y \in \mathcal{F}} \min_\beta \mathbb{E}_Q[(Y - \beta)\frac{dP}{dQ}] - \left\{ -\beta + \mathbb{E}_Q[\phi^*(Y)] \right\} \\
=&\max_{Y \in \mathcal{F}} \mathbb{E}_Q[Y\frac{dP}{dQ} - \phi^*(Y)] + \min_\beta \beta\mathbb{E}_Q[1 - \frac{dP}{dQ}] \\
=&\max_{Y \in \mathcal{F}} \mathbb{E}_Q[Y\frac{dP}{dQ} - \phi^*(Y)] + 0
\end{aligned}
\tag{37}
$$

The first equality follows from replacing $f + \theta_1$ with $Y$. The second equality is a re-arrangement for optimzing $\beta$. The third equation holds as $\mathbb{E}_Q[1 - \frac{dP}{dQ}] = 0$.

Applying Theorem A.3, the last supremum reduces to:

$$
\begin{aligned}
&\max_{Y \in \mathcal{F}} \mathbb{E}_Q[Y\frac{dP}{dQ} - \phi^*(Y)] \\
=&\mathbb{E}_Q[\sup_{Y \in \mathcal{F}} \{Y\frac{dP}{dQ} - \phi^*(Y)\}] \\
=&\mathbb{E}_Q[\phi^{**}(\frac{dP}{dQ})] \\
=&\mathbb{E}_Q[\phi(\frac{dP}{dQ})] \\
=&D_\phi(P||Q)
\end{aligned}
\tag{38}
$$

where the last equality follows from the fact that $\phi^{**} = \phi$. This concludes the proof.

$\square$

## A.6 Proof of $Q^*$

*Proof.* From theorm 3.2, CL-DRO can be rewrriten as:

$$
\begin{aligned}
\mathcal{L}_{\text{CL-DRO}}^\phi &= -\mathbb{E}_{P_0}[f_\theta] + \min_\beta \left\{ \alpha^*\eta - \beta + \alpha^*\mathbb{E}_{Q_0}[\max_L\{\frac{f_\theta + \beta}{\alpha^*}L - \phi(L)\}] \right\} \\
&= -\mathbb{E}_{P_0}[f_\theta] + \min_\beta \left\{ \alpha^*\eta - \beta + \alpha^*\mathbb{E}_{Q_0}[\phi^*(\frac{f_\theta + \beta}{\alpha^*})] \right\}
\end{aligned}
\tag{39}
$$

For the inner optimzation, we can draw the optimal $L$ for $\max_L\{\frac{f_\theta + \beta}{\alpha^*}L - \phi(L)\}$ as:

$$
L = e^{\frac{f_\theta + \beta}{\alpha^*}}
\tag{40}
$$

For the outer optimization, we can draw the optimal $\beta$ for $\min_\beta \left\{ \alpha^*\eta - \beta + \alpha^*\mathbb{E}_{Q_0}[e^{\frac{f_\theta + \beta}{\alpha^*}} - 1] \right\}$ as

$$
\beta = \alpha^* \log \mathbb{E}_{Q_0} e^{-\frac{f_\theta}{\alpha^*}}
\tag{41}
$$

Then we plug Equation (41) into Equation (40).

$$
L = \frac{e^{\frac{f_\theta}{\alpha^*}}}{\mathbb{E}_{Q_0}[e^{\frac{f_\theta}{\alpha^*}}]}
\tag{42}
$$

Based on the definition of $L$, we can derive the expression for $Q^*$:

$$Q^* = \frac{e^{\frac{f_\theta}{\alpha^*}}}{\mathbb{E}_{Q_0}[e^{\frac{f_\theta}{\alpha^*}}]} Q_0 \tag{43}$$

$\square$

## B  Experiments

Figure 5 shows PyTorch-style pseudocode for the standard objective, the adjusted InfoNCE objective. The proposed adjusted reweighting loss is very simple to implement, requiring only two extra lines of code compared to the standard objective.

```
1  # pos      : exp of inner products for positive examples
2  # neg      : exp of inner products for negative examples
3  # N        : number of negative examples
4  # t        : temperature scaling
5  # mu       : center position
6  # sigma    : height scale
7
8  #InfoNCE
9  standard_loss = -log(pos.sum() / (pos.sum() + neg.sum()))
10
11 #ADNCE
12 weight=1/(sigma * sqrt(2*pi)) * exp( -0.5 * ((neg-mu)/sigma)**2 )
13 weight=weight/weight.mean()
14 Adjusted_loss = -log(pos.sum() / (pos.sum() + (neg * weight.detach() ).sum())
       )
```

Figure 5: Pseudocode for our proposed adjusted InfoNCE objective, as well as the original NCE contrastive objective. The implementation of our adjusted reweighting method only requires two additional lines of code compared to the standard objective.

### B.1  Visual Representation

**Model.** For contrastive learning on images, we adopt SimCLR [4] as our baseline and follow the same experimental setup as [13]. Specifically, we use the ResNet-50 network as the backbone. To ensure a fair comparison, we set the embedded dimension to 2048 (the representation used in linear readout) and project it into a 128-dimensional space (the actual embedding used for contrastive learning). Regarding the temperature parameter $\tau$, we use the default value $\tau_0$ of 0.5 in most researches, and we also perform grid search on $\tau$ varying from 0.1 to 1.0 at an interval of 0.1, denoted by $\tau^*$. The best parameters for each dataset is reported in Table 6. Note that $\{\cdot\}$ indicates the range of hyperparameters that we tune and the numbers in **bold** are the final settings. For $\alpha$-CL, we follow the setting of [16], where $p = 4$ and $\tau = 0.5$. We use the Adam optimizer with a learning rate of 0.001 and weight decay of $1e - 6$. All models are trained for 400 epochs.

Table 6: hyperparameters setting on each datasets.

| DATASETS | CIFAR10 | STL10 | CIFAR100 |
|---|---|---|---|
| BEST $\tau$ | { 0.1, 0.2, 0.3, **0.4**, 0.5, 0.6 } | { 0.1, **0.2**, 0.3, 0.4, 0.5, 0.6 } | { 0.1, 0.2, **0.3**, 0.4, 0.5, 0.6 } |
| $\mu$ | { 0.5, 0.6, **0.7**, 0.8, 0.9 } | { 0.5, 0.6, 0.7, **0.8**, 0.9 } | **0.5**, 0.6, 0.7, 0.8, 0.9 } |
| $\sigma$ | {**0.5**, 1.0 } | {0.5, **1.0** } | {0.5, **1.0**} |

**Noisy experiments in Section 3.4.** To investigate the relationship between the temperature parameter $\tau$ (or $\eta$) and the noise ratio, we follow the approach outlined in [13] and utilize the class information of each image to select negative samples as a combination of true negative samples and false negative samples. Specifically, $r_{ratio} = 0$ indicates all negative samples are true negative samples, $r_{ratio} = 0.5$ suggests 50% of true positive samples existing in negative samples, $r_{ratio} = 1$ means uniform sampling.

**Variance analysis in Section 3.4.** To verify the mean-variance objective of InfoNCE, we adopt the approach outlined in [60] and record the negative prediction scores for 256 samples (assuming a batch size of 256) in each minibatch. Specifically, we randomly select samples from a batch to calculate the statistics and visualize them. (1) For positive samples, we calculate cosine similarity by taking the inner product after normalization, and retain the mean value of the 256 positive scores as '*pos mean*'. (2) For negative samples, we average the means and variances of 256 negative samples to show the statistical characteristics of these N negative samples '*(mean neg; var neg)*'. We record this data at each training step to track score distribution throughout the training process.

## B.2 Sentence Representation

For the sentence contrastive learning, we adopt the approach outlined in [9] and evaluate our method on 7 popular STS datasets: STS tasks from 2012-2016, STS-B and SICK-R. We utilize the SentEval toolkit to obtain all 7 datasets. Each dataset includes sentence pairs which are rated on a scale of 0 to 5, indicating the degree of semantic similarity. To validate the effective of our proposed method, we utilize several methods as baselines: average GloVe embeddings, BERT-flow, BERT-whitening, CT-BERT and SimCSE. The best parameters for each dataset is reported in Table 7. To ensure fairness, we employed the official code, which can be accessed at `https://github.com/princeton-nlp/SimCSE`.

Table 7: hyperparameters setting on sentence CL. Note that $\{\cdot\}$ indicates the range of hyperparameters that we tune and the numbers in **bold** are the final settings.

| DATASETS | SIMCSE-BERT$_{\text{BASE}}$ | SIMCSE-ROBERTA$_{\text{BASE}}$ |
|---|---|---|
| BEST $\tau$ | { 0.01, 0.02, 0.03, 0.04, 0.05, 0.06, **0.07**, 0.08, 0.09, 0.10, 0.15, 0.20} | { 0.01, 0.02, 0.03, 0.04, 0.05, **0.06**, 0.07, 0.08, 0.09, 0.10, 0.15, 0.20} |
| $\mu$ | {0.3, **0.4**, 0.5, 0.6, 0.7, 0.8, 0.9 } | { 0.5, 0.6, 0.7, 0.8, 0.9, 1.0, 1.5, **2.0**, 2.5, 3.0} |
| $\sigma$ | {0.5, **1.0** } | {0.5, **1.0**} |

## B.3 Graph Representation

For the graph contrastive learning experiments on TU-Dataset [61], we adopted the same experimental setup as outlined in [12]. The dataset statistics can be found in Table 8. To ensure fairness, we employed the official code, which can be accessed at `https://github.com/Shen-Lab/GraphCL/tree/master/unsupervised_TU`. We made only modifications to the script by incorporating our ADNCE method and conducting experiments on the hyper-parameter $\mu \in \{0.5, 0.6, 0.7, 0.8, 0.9, 1.0\}$ and $\sigma = 1$ on most datasets. Each parameter was repeated from scratch five times, and the best parameter was selected by evaluating on the validation dataset. The best parameters for each dataset is reported in Table 9.

We summarize the statistics of TU-datasets [61] for unsupervised learning in Table 8. Table 10 demonstrates the consistent superiority of our proposed ADNCE approach.

Table 8: Statistics for unsupervised learning TU-datasets.

| DATASETS | CATEGORY | GRAPHS# | AVG. N# | AVG. DEGREE |
|---|---|---|---|---|
| NCI1 | BIOCHEMICAL MOLECULES | 4,110 | 29.87 | 1.08 |
| PROTEINS | BIOCHEMICAL MOLECULES | 1,113 | 39.06 | 1.86 |
| DD | BIOCHEMICAL MOLECULES | 1,178 | 284.32 | 715.66 |
| MUTAG | BIOCHEMICAL MOLECULES | 188 | 17.93 | 19.79 |
| COLLAB | SOCIAL NETWORKS | 5,000 | 74.49 | 32.99 |
| RDT-B | SOCIAL NETWORKS | 2,000 | 429.63 | 1.15 |
| RDT-M | SOCIAL NETWORKS | 2,000 | 429.63 | 497.75 |
| IMDB-B | SOCIAL NETWORKS | 1,000 | 19.77 | 96.53 |

Table 9: hyperparameters setting on graph CL. Note that $\{\cdot\}$ indicates the range of hyperparameters that we tune and the numbers in **bold** are the final settings.

| Datasets | Best $\tau$ | $\mu$ | $\sigma$ |
|---|---|---|---|
| NCI1 | { **0.05**, 0.10, 0.15, 0.20, 0.25} | { 0.2, 0.3 ,0.4, 0.5, 0.6, **0.7**, 0.8, 0.9 } | { 0.5, **1.0** } |
| PROTEINS | { **0.05**, 0.10, 0.15, 0.20, 0.25} | { 0.5, 1.0, **1.5**, 2.0 } | { 0.5, **1.0** } |
| DD | { 0.05, 0.10, 0.15, **0.20**, 0.25} | { **0.2**, 0.3 ,0.4, 0.5, 0.6, 0.7, 0.8, 0.9 } | { 0.5, **1.0** } |
| MUTAG | { 0.05, 0.10, **0.15**, 0.20, 0.25} | { 0.2, 0.3 ,0.4, 0.5, 0.6, **0.7**, 0.8, 0.9 } | { 0.5, **1.0** } |
| COLLAB | { 0.05, **0.10**, 0.15, 0.20, 0.25} | { **0.2**, 0.3 ,0.4, 0.5, 0.6, 0.7, 0.8, 0.9 } | { 0.5, **1.0** } |
| RDT-B | { 0.05, 0.10, **0.15**, 0.20, 0.25} | { 0.2, **0.3** ,0.4, 0.5, 0.6, 0.7, 0.8, 0.9 } | { 0.5, **1.0** } |
| RDT-M | { 0.05, 0.10, **0.15**, 0.20, 0.25} | { 0.2, 0.3 ,0.4, 0.5, 0.6, 0.7, **0.8**, 0.9 } | { 0.5, **1.0** } |
| IMDB-B | { 0.10, 0.20, 0.30, 0.40, **0.50**} | { 0.2, 0.3 ,0.4, 0.5, 0.6, **0.7**, 0.8, 0.9 } | { 0.5, **1.0** } |

Table 10: Unsupervised representation learning classification accuracy (%) on TU datasets. The compared numbers are from except AD-GCL, whose statistics are reproduced on our platform. **Bold** indicates the best performance while underline indicates the second best on each dataset.

| Dataset | NCI1 | PROTEINS | DD | MUTAG | COLLAB | RDT-B | RDT-M5K | IMDB-B | AVG. |
|---|---|---|---|---|---|---|---|---|---|
| No Pre-Train | 65.40±0.17 | 72.73±0.51 | 75.67±0.29 | 87.39±1.09 | 65.29±0.16 | 76.86 ±0.25 | 48.48±0.28 | 69.37±0.37 | 70.15 |
| InfoGraph | 76.20± 1.06 | 74.44± 0.31 | 72.85± 1.78 | 89.01±1.13 | 70.05±1.13 | 82.50±1.42 | 53.46±1.03 | 73.03±0.87 | 74.02 |
| GraphCL | 77.87±0.41 | 74.39±0.45 | 78.62±0.40 | 86.80±1.34 | 71.36±1.15 | 89.53±0.84 | 55.99±0.28 | 71.14±0.44 | 75.71 |
| AD-GCL | 73.91±0.77 | 73.28±0.46 | 75.79±0.87 | 88.74±1.85 | 72.02±0.56 | 90.07±0.85 | 54.33±0.32 | 70.21±0.68 | 74.79 |
| RGCL | 78.14±1.08 | 75.03±0.43 | 78.86±0.48 | 87.66±1.01 | 70.92±0.65 | 90.34±0.58 | **56.38±0.40** | **71.85±0.84** | 76.15 |
| ADNCE | **79.30±0.67** | **75.10±0.25** | **79.23±0.59** | **89.04±1.30** | **72.26±1.10** | **91.39±0.31** | 56.01±0.35 | 71.58±0.72 | **76.74** |

## B.4 Additional ablation study

The primary motivation behind ADNCE is to mitigate the issues of over-conservatism and sensitivity to outliers. These limitations stem from the unreasonable worst-case distribution, which assigns excessive weights to the hardest negative samples. Consequently, any weighting strategy capable of modulating the worst-case distribution to focus more on the informative region holds promising. In our ADNCE, we opted for Gaussian-like weights due to its flexibility and unimodal nature. However, alternative weighting strategies such as Gamma, Rayleigh or Chi-squared could also be employed. The following experiment demonstrates that these alternative weighting strategies can yield comparable results to Gaussian-like weights. The above table showcases the comparison of the

Table 11: Alternative weighting strategies for ADNCE.

| Weight Strategy | Probability density function | CIFAR10 |
|---|---|---|
| Gamma-like | $w(x, m, n) = \frac{1}{\Gamma(m)n^m} x^{m-1} e^{-\frac{x}{n}}$ | 91.74 |
| Rayleigh-like | $w(x, m) = \frac{x}{m^2} e^{-\frac{x^2}{2m^2}}$ | 91.73 |
| Chi-squared-like | $w(x, m) = \frac{1}{2^{m/2}\Gamma(m/2)} x^{m/2-1} e^{-x/2}$ | 91.99 |
| Gaussian-like (ADNCE) | $w(x, m, n) = \frac{1}{n\sqrt{2\pi}} e^{-\frac{1}{2}(\frac{x-m}{n})^2}$ | 91.88 |

TOP-1 accuracy performance on CIFAR10 dataset under different weight strategies. The parameters $m$ and $n$ are utilized to denote the parameters within their corresponding probability density function, while the variable $x$ represents a random variable. It is important to note that, due to the domain definition of some PDFs being $(0, +\infty)$, we need to set $x = prediction\_score + 1$.

Table 12: Sensitivity analysis of $\mu$.

| $\mu$ | 0.1 | 0.3 | 0.5 | 0.7 | 0.9 |
|---|---|---|---|---|---|
| CIFAR10 | 91.3 | 91.66 | 91.9 | 91.77 | 92.25 |
| STL10 | 87.84 | 88.22 | 87.56 | 88.48 | 88.45 |
| CIFAR100 | 69.34 | 69.31 | 68.70 | 69.24 | 68.95 |

Table 13: Sensitivity analysis of $\sigma$.

| $\sigma$ | 0.2 | 0.4 | 0.6 | 0.8 | 1 | 1.5 | 2 |
|---|---|---|---|---|---|---|---|
| CIFAR10 | 90.07 | 91.85 | 92.02 | 91.77 | 91.72 | 91.69 | 91.94 |
| STL10 | 86.54 | 88.30 | 88.10 | 87.54 | 88.95 | 88.12 | 88.40 |
| CIFAR100 | 67.38 | 69.36 | 69.52 | 69.01 | 68.70 | 69.24 | 69.42 |

## B.5 Sensitivity analysis of $\mu$ and $\sigma$

The above tables showcase the comparisons of the TOP-1 accuracy performance on three dataset under different $\mu$ and $\sigma$. As can be seen, changing the parameters $\mu$ and $\sigma$ would impact the model performance, but not as dramatical as tuning the parameter $\tau$. (For example, on STL10 $\sigma$ from 1.0 to 0.2 just brings 2.7% performance drops, while changing $\tau$ brings 7.6% performance gap.) This outcome indicates that tuning $\mu$ and $\sigma$ is not a significant burden. In most scenarios, it may suffice to set $\sigma = 1$, requiring only the tuning of $\mu$ within the range of 0.1 to 0.9 (can refer to Table 6, 7, 9 in Appendix B for more details).

