# A  Appendix of Proofs

## A.1  Proof of Thm.3.2

**Theorem 3.2.** *By choosing KL divergence $D_{KL}(Q||Q_0) = \int Q \log \frac{Q}{Q_0} dx$, optimizing CL-DRO (cf. Eqn. 3) is equivalent to optimizing CL (InfoNCE,cf. Eqn. 1):*

$$
\begin{aligned}
\mathcal{L}_{CL\text{-}DRO}^{KL} &= -\mathbb{E}_{P_0}[f_\theta] + \min_{\alpha \geq 0, \eta_1} \max_{Q \in \mathbb{Q}} \{\mathbb{E}_Q[f_\theta] - \alpha[D_{KL}(Q||Q_0) - \eta] + \eta_1(\mathbb{E}_{Q_0}[\frac{Q}{Q_0}] - 1)\} \\
&= -\mathbb{E}_{P_0}\left[\alpha^* \log \frac{e^{f_\theta/\alpha^*}}{\mathbb{E}_{Q_0}[e^{f_\theta/\alpha^*}]}\right] + Constant = \alpha^* \mathcal{L}_{InfoNCE} + Constant
\end{aligned}
\tag{4}
$$

*where $\alpha, \eta_1$ represent the Lagrange multipliers, and the optimal $\alpha^*$ finally serves as the temperature $\tau$ in CL.*

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

{1}{N-1+\exp\left(\frac{1}{\tau}\right)}\sqrt{\frac{N\exp\left(\frac{2}{\tau}\right)\log\left(\frac{1}{\rho}\right)}{2}}$.*

Here we simply disregard the constant term present in Eqn. 4 as it does not impact optimization, and omit the error from the positive instances.

*Proof.* Before detailing the proof process, we first introduce a pertinent theorem:

**Theorem A.4** (McDiarmid's inequality [10]). *Let $X_1, \cdots, X_n$ be independent random variables, where $X_i$ has range $\mathcal{X}$. Let $f : \mathcal{X}_1 \times \cdots \times \mathcal{X}_n \to \mathbb{R}$ be any function with the $(c_1, \ldots, c_n)$-bounded difference property: for every $i = 1, \ldots, n$ and every $(x_1, \ldots, x_n), (x_1', \ldots, x_n') \in \mathcal{X}_1 \times \cdots \times \mathcal{X}_n$ that differ only in the $i$-th coordinate ($x_j = x_j'$ for all $j \neq i$), we have $|f(x_1, \ldots, x_n) - f(x_1', \ldots, x_n')| \leq c_i$. For any $\epsilon > 0$,*

$$
\mathbb{P}(f(X_1, \cdots, X_n) - \mathbb{E}[f(X_1, \cdots, X_n)] \geq \epsilon) \leq \exp\left(\frac{-2\epsilon^2}{\sum_{i=1}^N c_i^2}\right)
\tag{19}
$$

Now we delve into the proof. As $Q^{ideal}$ satisfies $D_{KL}(Q||Q_0) \leq \eta$, we can bound $\mathcal{L}_{unbiased}$ with:

$$
\begin{aligned}
\mathcal{L}_{unbiased} &= -\mathbb{E}_{P_0}[f_\theta] + \mathbb{E}_{Q^{ideal}}[f_\theta] \\
&\leq -\mathbb{E}_{P_0}[f_\theta] + \max_{D_{KL}(Q|Q_0)\leq\eta}\mathbb{E}_Q[f_\theta] \\
&= \mathcal{L}_{\text{CL-DRO}}^{KL}
\end{aligned}
\tag{20}
$$

where $Q^{ideal}, Q^*$ denotes the ideal negative distribution and the worst-case distribution in CL-DRO. From the Thm.3.2, we have the equivalence between InfoNCE and CL-DRO. Thus here we choose CL-DRO for analyses. Suppose we have $N$ negative samples, and for any pair of samples $(x_i, y_i), (x_j, y_j)$, we have the following bound:

$$
|Q^*(x_i, y_i)f_\theta(x_i, y_i) - Q^*(x_j, y_j)f_\theta(x_j, y_j)| \leq \sup_{(x,y)\sim Q_0}|Q^*(x, y)f_\theta(x, y)| \leq \frac{\exp\left(\frac{1}{\tau}\right)}{N - 1 + \exp\left(\frac{1}{\tau}\right)}
\tag{21}
$$

where the first inequality holds as $Q^*(x, y)f_\theta(x, y) > 0$. The second inequality holds based on the expression of $Q^* = Q_0\frac{\exp[f_\theta/\tau]}{E_{Q_0}\exp[f_\theta/\tau]}$ (refer to Appendix A.6). Suppose $f_\theta \in [M_1, M_2]$, the

49   upper bound of $\sup_{(x,y)\sim Q_0} |Q^*(x,y)f_\theta(x,y)|$ arrives if $f_\theta(x,y) = M_2$ for the sample $(x,y)$ and

50   $f_\theta(x,y) = M_1$ for others. We have $\sup_{(x,y)\sim Q_0} |Q^*(x,y)f_\theta(x,y)| \leq \frac{M_2 \exp((M_2-M_1)/\tau)}{N-1+\exp((M_2-M_1)/\tau)}$. In

51   this work, for brevity, here we simply consider $M_1 = 0, M_2 = 1$ for analyses. It shares the common

52   properties with the general interval $[M_1, M_2]$.

53   By using McDiarmid's inequality in Thm A.4, for any $\epsilon$, we have:

$$\mathbb{P}[(\mathcal{L}_{\text{CL-DRO}}^{KL} - \tau\widehat{\mathcal{L}}_{\text{InfoNCE}}) \geq \epsilon]$$
$$\leq \exp\Big(\frac{-2\epsilon^2(N-1+\exp(\frac{1}{\tau}))^2}{N\exp(\frac{2}{\tau})}\Big) \tag{22}$$

54   Let

$$\rho = \exp(\frac{-2\epsilon^2(N-1+\exp(\frac{1}{\tau}))^2}{N\exp(\frac{2}{\tau})}) \tag{23}$$

55   we get:

$$\epsilon = \frac{1}{N-1+\exp\left(\frac{1}{\tau}\right)}\sqrt{\frac{N\exp\left(\frac{2}{\tau}\right)\log\left(\frac{1}{\rho}\right)}{2}} \tag{24}$$

56   Thus, for $\forall \rho \in (0,1)$, we conclude that with probability at least $1 - \rho$.

$$\mathcal{L}_{\text{unbiased}} \leq \widehat{\mathcal{L}}_{\text{InfoNCE}} + \frac{1}{N-1+\exp\left(\frac{1}{\tau}\right)}\sqrt{\frac{N\exp\left(\frac{2}{\tau}\right)\log\left(\frac{1}{\rho}\right)}{2}} \tag{25}$$

57   $\square$

## A.3   Proof of Coro.3.4

59   **Corollary 3.4.** *[The optimal $\alpha$ - Lemma 5 of [6]] The value of the optimal $\alpha$ (i.e., $\tau$) can be*
60   *approximated as follow:*

$$\tau \approx \sqrt{\mathbb{V}_{Q_0}[f_\theta]/2\eta}. \tag{6}$$

61   *where $\mathbb{V}_{Q_0}[f_\theta]$ denotes the variance of $f_\theta$ under the distribution $Q_0$.*

62   *Proof.* While Corollary 3.4 has already been proven in [6], we present a brief outline of the proof here
63   for the sake of completeness and to ensure that our article is self-contained. To verify the relationship
64   between $\tau$ and $\eta$, we could utilize the approximate expression of InfoNCE (*cf.* Eqn. 29) and focus on
65   the first order conditions for $\tau$. In detail, we have:

$$-\mathbb{E}_{P_0}[f_\theta] + \inf_{\alpha\geq 0}\{\mathbb{E}_{Q_0}[f_\theta] - \frac{1}{2\alpha}\frac{1}{\phi^{(2)}(1)}\mathbb{V}_{Q_0}[f_\theta] - \alpha\eta\}$$

66   To find the optimal value of $\alpha$ (or equivalently, $\tau$), we differentiate the above equation and set it to 0.
67   This yields a fixed-point equation

$$\tau = \sqrt{\frac{\mathbb{V}_{Q_0}[f_\theta]}{2\eta}}$$

68   The corollary gets proved.   $\square$

## A.4   Proof of Thm.3.5

70   **Theorem 3.5.** *Given any $\phi$-divergence, the corresponding CL-DRO objective could be approximated*
71   *as a mean-variance objective:*

$$\mathcal{L}_{CL\text{-}DRO}^{\phi}(f_\theta) \approx -\mathbb{E}_{P_0}[f_\theta] + (\mathbb{E}_{Q_0}[f_\theta] + \frac{1}{2\tau}\frac{1}{\phi^{(2)}(1)}\cdot\mathbb{V}_{Q_0}[f_\theta]) \tag{7}$$

72   *where $\phi^{(2)}(1)$ denotes the the second derivative value of $\phi(\cdot)$ at point 1, and $\mathbb{V}_{Q_0}[f_\theta]$ denotes the*
73   *variance of $f$ under the distribution $Q_0$.*

74   *Specially, if we consider KL divergence, the approximation transforms:*

$$\mathcal{L}_{CL\text{-}DRO}^{KL}(f_\theta) \approx -\mathbb{E}_{P_0}[f_\theta] + (\mathbb{E}_{Q_0}[f_\theta] + \frac{1}{2\tau}\mathbb{V}_{Q_0}[f_\theta]) \tag{8}$$

 *Proof.* We start with introducing a useful lemma.

 **Lemma A.5** (Lemma A.2 of [8]). *Suppose that $\phi : \mathbb{R} \to \mathbb{R} \bigcup \{+\infty\}$ is a closed, convex function*
 *such that $\phi(z) \geq \phi(1) = 0$ for all $z$, is two times continuously differentiable around $z = 1$, and*
 *$\phi(1) > 0$, Then*

$$
\begin{aligned}
\phi^*(\zeta) &= \max_z \{z\zeta - \phi(z)\} \\
&= \zeta + \frac{1}{2!}[\frac{1}{\phi''(1)}]\zeta^2 + o(\zeta^2)
\end{aligned}
\tag{26}
$$

 Note that most of the $\phi$-divergences [13] (*e.g.,* KL divergence, Cressie-Read divergence, Burg entropy,
 J-divergence, $\chi^2$-distance, modified $\chi^2$-distance, and Hellinger distance) satisfy the smoothness
 conditions. When $n = 2$, $\phi^*[\zeta] \approx \zeta + \frac{1}{2}[\frac{1}{\phi^{(2)}(1)}]\zeta^2$. Substituting this back to Eqn.17 we have:

$$
\begin{aligned}
\mathcal{L}_{\text{CL-DRO}}^\phi &= -\mathbb{E}_{P_0}[f_\theta] + \min_{\alpha \geq 0, \eta_1} \left\{ \alpha\eta - \eta_1 + \alpha\mathbb{E}_{Q_0}[\phi^*(\frac{f_\theta + \eta_1}{\alpha})] \right\} \\
&= -\mathbb{E}_{P_0}[f_\theta] + \min_{\alpha \geq 0, \eta_1} \left\{ \alpha\eta - \eta_1 + \alpha\mathbb{E}_{Q_0}[\frac{f_\theta + \eta_1}{\alpha} + \frac{1}{2}\frac{1}{\phi^{(2)}(1)}(\frac{f_\theta + \eta_1}{\alpha})^2] \right\} \\
&= -\mathbb{E}_{P_0}[f_\theta] + \min_{\alpha \geq 0, \eta_1} \left\{ \alpha\eta - \eta_1 + \mathbb{E}_{Q_0}[f_\theta + \eta_1 + \frac{1}{2}\frac{1}{\phi^{(2)}(1)\alpha}(f_\theta + \eta_1)^2] \right\} \\
&= -\mathbb{E}_{P_0}[f_\theta] + \min_{\alpha \geq 0, \eta_1} \left\{ \alpha\eta + \mathbb{E}_{Q_0}[f_\theta + \frac{1}{2}\frac{1}{\phi^{(2)}(1)\alpha}(f_\theta + \eta_1)^2] \right\}
\end{aligned}
\tag{27}
$$

 If we differentiate it *w.r.t.* $\eta_1$:

$$
\frac{\partial\{\alpha\eta + \mathbb{E}_{Q_0}[f_\theta + \frac{1}{2}\frac{1}{\phi^{(2)}(1)\alpha}(f_\theta + \eta_1)^2]\}}{\partial\eta_1} = 0
\tag{28}
$$

 we have $\eta_1^* = -\mathbb{E}_{Q_0}[f_\theta]$, and the objective transforms into:

$$
\begin{aligned}
&-\mathbb{E}_{P_0}[f_\theta] + \inf_{\alpha \geq 0, \eta_1} \left\{ \alpha\eta + \mathbb{E}_{Q_0}[f_\theta + \frac{1}{2}\frac{1}{\phi^{(2)}(1)\alpha}(f_\theta + \eta_1)^2] \right\} \\
&= -\mathbb{E}_{P_0}[f_\theta] + \inf_{\alpha \geq 0} \{\mathbb{E}_{Q_0}[f_\theta + \frac{1}{2\alpha}\frac{1}{\phi^{(2)}(1)}(f_\theta - \mathbb{E}_{Q_0}[f_\theta])^2] - \alpha\eta\} \\
&= -\mathbb{E}_{P_0}[f_\theta] + \inf_{\alpha \geq 0} \{\mathbb{E}_{Q_0}[f_\theta] + \frac{1}{2\alpha}\frac{1}{\phi^{(2)}(1)}\mathbb{V}_{Q_0}[f_\theta] - \alpha\eta\}
\end{aligned}
\tag{29}
$$

 Choosing KL-divergence, we have $\phi^{(2)}(1) = 1$. Substituting $\alpha^*(\tau)$ into Eqn. 29 and ignoring the
 constant $\alpha\eta$:

$$
-\mathbb{E}_{P_0}[f_\theta] + \mathbb{E}_{Q_0}[f_\theta] + \frac{1}{2\tau}\mathbb{V}_{Q_0}[f_\theta]
$$

 Then Theorem gets proved. $\qquad\square$

 ## A.5 Proof of Thm.4.2

 **Theorem 4.2.** *For distributions $P, Q$ such that $P \ll Q$, let $\mathcal{F}$ be a set of bounded measurable*
 *functions. Let CL-DRO draw positive and negative instances from $P$ and $Q$, marked as $\mathcal{L}_{CL\text{-}DRO}^\phi(P, Q)$.*
 *Then the CL-DRO objective is the tight variational estimation of $\phi$-divergence. In fact, we have:*

$$
D_\phi(P||Q) = \sup_{f \in \mathcal{F}} -\mathcal{L}_{CL\text{-}DRO}^\phi(P, Q) = \sup_{f \in \mathcal{F}} \mathbb{E}_P[f] - \min_{\lambda \in \mathbb{R}} \{\lambda + \mathbb{E}_Q[\phi^*(f - \lambda)]\}
\tag{10}
$$

 *Here, the choice of $\phi$ in CL-DRO corresponds to the probability measures in $D_\phi(P||Q)$.*

 *Proof.* Regarding this theorem, our proof primarily relies on the variational representation of $\phi$-
 divergence and optimized certainty equivalent (OCE) risk. Towards this end, we start to introduce the
 basic concepts:

**Definition A.6** (OCE [2]). Let $X$ be a random variable and let $u$ be a convex, lower-semicontinuous function satisfies $u(0) = 0, u^*(1) = 0$, then optimized certainty equivalent (OCE) risk $\rho(X)$ is defines as:

$$\rho(X) = \inf_{\lambda \in \mathbb{R}} \{\lambda + \mathbb{E}[u(f - \lambda)]\} \tag{30}$$

OCE is a type of risk measure that is widely used by both practitioners and academics [1, 2]. With duality theory, its various properties have been inspiring in our study of DRO.

**Definition A.7** (Variational formulation).

$$D_\phi(P||Q) := \sup_{f \in \mathcal{F}} \{\mathbb{E}_P[f] - \mathbb{E}_Q[\phi^*(f)]\} \tag{31}$$

where the supremum is taken over all bounded real-valued measurable functions $\mathcal{F}$ defined on $\mathcal{X}$.

Note that in order to keep consistent with the definition of CL-DRO, we transform Eqn.10 to :

$$D_\phi(P||Q) = \sup_{f \in \mathcal{F}} -\mathcal{L}_{\text{CL-DRO}}^\phi(P, Q) = \sup_{f \in \mathcal{F}} \mathbb{E}_P[f] - \inf_{\eta_1 \in \mathbb{R}} \{-\eta_1 + \mathbb{E}_Q[\phi^*(f + \eta_1)]\} \tag{32}$$

Our proof for this theorem primarily relies on utilizing OCE risk as a bridge and can be divided into two distinct steps:

Step 1: $\sup_{f \in \mathcal{F}} -\mathcal{L}_{\text{CL-DRO}}^\phi(P, Q) = \sup_{f \in \mathcal{F}} \mathbb{E}_P[f] - \inf_{\eta_1 \in \mathbb{R}} \{-\eta_1 + \mathbb{E}_Q[\phi^*(f + \eta_1)]\}$.

Step 2: $D_\phi(P||Q) = \sup_{f \in \mathcal{F}} \mathbb{E}_P[f] - \inf_{\eta_1 \in \mathbb{R}} \{-\eta_1 + \mathbb{E}_Q[\phi^*(f + \eta_1)]\}$

1. **We show that** $\sup_{f \in \mathcal{F}} -\mathcal{L}_{\textbf{CL-DRO}}^\phi(P, Q) = \sup_{f \in \mathcal{F}} \mathbb{E}_P[f] - \inf_{\eta_1 \in \mathbb{R}} \{-\eta_1 + \mathbb{E}_Q[\phi^*(f + \eta_1)]\}$.

$$
\begin{aligned}
-\mathcal{L}_{\text{CL-DRO}}^\phi &= \mathbb{E}_P[f] - \inf_{\alpha \geq 0, \eta_1} \sup_L \{\mathbb{E}_Q[fL] - \alpha[\mathbb{E}_Q[\phi(L)] - \eta] + \eta_1(\mathbb{E}_Q[L] - 1)\} \\
&= \mathbb{E}_P[f] - \inf_{\alpha \geq 0, \eta_1} \{\alpha\eta - \eta_1 + \alpha\mathbb{E}_Q[\phi^*(\frac{f + \eta_1}{\alpha})]\} \\
&= \mathbb{E}_P[f] - \inf_{\eta_1} \{\alpha^*\eta - \eta_1 + \alpha^*\mathbb{E}_Q[\phi^*(\frac{f + \eta_1}{\alpha^*})]\} \\
&= \mathbb{E}_P[f] - \inf_{\eta_1} \{-\eta_1 + \alpha^*\mathbb{E}_Q[\phi^*(\frac{f + \eta_1}{\alpha^*})] + Constant\}
\end{aligned} \tag{33}
$$

When $\alpha^* = 1$, step 1 gets proved.

2. **We show that** $D_\phi(P||Q) = \sup_{f \in \mathcal{F}} \mathbb{E}_P[f] - \inf_{\eta_1 \in \mathbb{R}} \{-\eta_1 + \mathbb{E}_Q[\phi^*(f + \eta_1)]\}$.

Firstly, we transform $\mathbb{E}_P[f]$ to $\mathbb{E}_Q[f \frac{dP}{dQ}]$ as:

$$
\begin{aligned}
&\sup_{f \in \mathcal{F}} \mathbb{E}_P[f] - \inf_{\eta_1} \{-\eta_1 + \mathbb{E}_Q[\phi^*(f + \eta_1)]\} \\
&= \sup_{f \in \mathcal{F}} \mathbb{E}_Q[f \frac{dP}{dQ}] - \inf_{\eta_1} \{-\eta_1 + \mathbb{E}_Q[\phi^*(f + \eta_1)]\}
\end{aligned} \tag{34}
$$

Let $f + \eta_1 = Y$, we have:

$$
\begin{aligned}
&\sup_{f \in \mathcal{F}} \mathbb{E}_Q[f \frac{dP}{dQ}] - \inf_{\eta_1} \{-\eta_1 + \mathbb{E}_Q[\phi^*(f + \eta_1)]\} \\
&= \sup_{Y \in \mathcal{F}} \inf_{\eta_1} \mathbb{E}_Q[(Y - \eta_1)\frac{dP}{dQ}] - \{-\eta_1 + \mathbb{E}_Q[\phi^*(Y)]\} \\