# OpenReview forum: "Understanding Contrastive Learning via Distributionally Robust Optimization"
_NeurIPS.cc/2023/Conference — NeurIPS 2023 poster_

### Official Review · Reviewer_C48x · 2023-06-15

**Soundness:** 2 fair
**Presentation:** 3 good
**Contribution:** 2 fair
**Rating:** 5
**Confidence:** 3

**Summary:**

This paper tackles the sampling bias problem in contrastive learning, where negative samples are usually sampled randomly from the marginal distribution and may contain false positive samples.
The authors provide a connection between the contrastive learning objective and distributionally robust optimization as a first part of the contributions:
By an optimal choice of the temperature parameter, the InfoNCE loss coincides with maximizing the loss for negative samples, leading to the worst-case negative sample distribution.
Consequently, it is revealed that the temperature parameter admits a trade-off between the generalization error and the DRO radius.
Secondly, the authors point out the two issues of standard contrastive learning:
It tends to overweigh negative samples with large distances (called over-conservatism) and be sensitive to outliers.
To mitigate these issues, the authors propose an alternative weight distribution over negative pairs to prevent overweighting of too-far negative samples.

**Strengths:**

- New insights on contrastive learning: One of the main contributions of this paper, the equivalent reformulation of contrastive learning with distributionally robust optimization, provides a new perspective to contrastive learning. In this picture, contrastive learning can be interpreted as optimizing the worst-case distribution over the negative samples. This interpretation can benefit us in developing a new method, as seen in Section 5.
- An new interpretation of the temperature parameter: As a result of the reformulation, we observe that the temperature parameter in contrastive learning serves as the Lagrange multiplier for the DRO constraint, and lower temperature leads to stronger DRO constraint. This insight again contributes to the development of algorithms.

**Weaknesses:**

- Meaning of the connection between contrastive learning and DRO is not clear enough: Theorem 3.2 connects the DRO objective and the InfoNCE loss. Although it is very interesting and insightful, and the authors argue that contrastive learning can be regarded as DRO and thus mitigate the sampling bias, we could argue that DRO is merely contrastive learning with the sampling bias in the opposite way. This would undermine the robustness property of DRO can still suffer from the sampling bias residing in contrastive learning.
- DRO interpretation may not explain hard mining: In Section 3.3.C and Section 5.1, the authors observe that the DRO equivalent formulation explains how contrastive learning puts larger weights to negative samples with larger $f\_\\theta$. Although this seems correct, this phenomenon does not correspond to what is so-called "hard mining." Indeed, Robinson et al. [38] (and many other works) define hard negative samples as "points that are difficult to distinguish from an anchor point," which can be put in the current context as "negative samples with small $f\_\\theta$." This observation may explain why a "strange" trade-off is observed in Section 3.3.A such that stronger DRO constraint leads to looser generalization error, unlike what we expect.

**Questions:**

Major comments are listed first.

- In Corollary 3.4 and Theorem 3.5, it is better to explain exactly what the "approximation" indicates.
- With the proposed method ADNCE, can you discuss how we should choose the parameters $\\mu$ and $\\sigma$ in Section 5.2?

Minor comments follow.

- Typo in l.42: "we proof" -> "we prove"
- Typo in l.82: "Donsker-Varadah" -> "Donsker-Varadhan"
- Typo in footnote 1: The expectation looks strange.
- Typo in l.142: $\\mathbb{E}\_Q$ -> $\\mathbb{E}\_{Q\_0}$
- In Theorem 3.2, the definition of $\\mathbb{Q}$ lacks. In Eq. (4), can $\\mathbb{E}\_{Q\_0}[Q/Q\_0]$ be simply written as $\\int Qdx$?
- Typo in l.156: "satisfied" -> "satisfies"
- Typo in l.173: "theorem 3.2" -> "Theorem 3.2"
- Typo in l.177: "subsection 3.4" -> "Subsection 3.4"
- Typo in l.190 and l.191: $E\_{Q\_0}$ -> $\\mathbb{E}\_{Q\_0}$
- In Theorem 4.2, it is better to mention $\\phi^\*$ indicates the convex conjugate because this is the first place where it appears.
- In Corollary 4.3, explaining what two random variables $X$ and $Y$ refer to make the statement complete is better.
- In l.251, what do you compare CL-DRO with by the word "tighter"? By the way, the word "tight" indicates that equality can be attained for a given inequality, so the comparative "tighter" seems strange.
- In l.257, I don't understand how "DRO bridges the gap" in the following paragraph.
- In l.259, it is better to explain what $\\mathcal{I}\_{DV}$ indicates.

**Limitations:**

The authors have adequately discussed the limitations in the conclusion.

Societal impacts do not apply to this work because this work is mostly theoretical.

---

> ### Author Rebuttal · Authors · 2023-08-09
>
> # Response to Reviewer C48x:
>
> Dear Reviewer,
>
> Much thanks for your detailed comments. In the revised version, we will meticulously polish the paper in accordance with your feedback, correcting typos and providing clear explanations of notations. Also, we find there may exist some misunderstandings.
> ## We would like to make clarifications in the following:
>
> **M1: The meaning of the connection between CL and DRO is not clear enough: ... We could argue that DRO is merely contrastive learning with the sampling bias in the opposite way. This would undermine the robustness property of DRO can still suffer from the sampling bias residing in CL.**
>
> A1: It appears there exists a misunderstanding. It is not accurate to consider DRO as merely contrastive learning with the sampling bias. DRO, in essence, is a broad optimization framework aiming to minimize the worst-case expected loss across a range of potential distributions. Our work (Thm. 3.2) demonstrates that optimizing the InfoNCE loss is equivalent to optimizing a CL-DRO objective, which is a specific loss function utilizing DRO. However, it is not logically sound to extrapolate this specific instance to conclude that DRO in general is equivalent to CL with sampling bias.
>
> Furthermore, the issue of sampling bias arises from the inadequate specification of the negative sampling distribution $Q\_0$. DRO itself does not involve how to specify $Q\_0$. Hence, it is not valid to deduce that sampling bias is intrinsic to DRO. In fact, the adversarial nature of DRO equips CL with robustness to sampling bias.
>
> **M2: DRO interpretation may not explain hard mining: ...  Indeed, Robinson et al. [38] (and many other works) define hard negative samples as "points that are difficult to distinguish from an anchor point," which can be put in the current context as "negative samples with small $f\_\theta$.**
>
> A2: There seems to be a misunderstanding that we would like to clarify. Our definition of hard negative samples aligns with the definition provided in [38]. More specifically, $f\_\theta$ in our paper represents the similarity between two points (as stated on line 98) --- a larger $f\_\theta$ value indicates a higher degree of similarity. Therefore, negative samples with large $f\_\theta$ values are considered as hard negative samples, not those with small $f\_\theta$ values.
>
> Regarding the scenario with an overly stringent DRO constraint, the decline in generalization error can be intuitively understood. In such cases, the optimization process takes into account an excessively broad range of distributions, including those irrelevant or harmful ones that significantly deviate from the ideal distribution. This can negatively impact the performance of the model.
>
> ## We further provide responses to the questions you have raised:
>
> **Q3: What the "approximation" indicates in Corollary 3.4 and Theorem 3.5?**
>
> A3: The term "approximation" refers to the estimation obtained by employing a second-order Taylor expansion on the objective function $\mathcal{L}\_{\text{CL-DRO}}^{\phi}$. By selecting the form of the Kullback-Leibler (KL) divergence as $\phi^{(2)}(1)=1$, truncating it to the second order, and differentiating it with respect to $\eta\_1$, we can derive $\mathcal{L}\_{\text{CL-DRO,2}}^{KL}$.
>
> Regarding the approximation of corollary 3.4, it is sufficient to differentiate $\mathcal{L}\_{\text{CL-DRO,2}}^{KL}$ with respect to $\alpha$ in order to obtain the final result.
>
> The reviewer could refer to Appendix A.3 and Appendix A.4 for more details.
>
> **Q4: How we should choose the  $\mu$ and $\sigma$?**
>
> A4: In most datasets, it is sufficient to set $\sigma=1$ and just needs to tune the hyperparameter $\mu$ via grid search. The reviewer could refer to Tables 6, 7, and 9 in Appendix B for more details.
>
> **Q5: What do you compare CL-DRO with by the word "tighter"?**
>
> A5: Existing common variational approximation of $\phi$-divergences is Donsker-Varadhan target ($I_{DV}$) : $D\_{\phi}(P||Q) \coloneqq \operatorname*{sup}\_{{f}\in \mathcal{F}} \{ \mathbb{E}\_P [f ] -  \mathbb{E}\_Q [\phi^*(f)] \}$
> which holds for an arbitrary finite measure in $\mathcal{F}$—is loose when applied to probability measures as was first observed in Ruderman et al. [1]. This expression fails to take into account the fact that divergences are defined between probability distributions. If we additionally assume, as we do for $\mathbb{R}$, that $\mathbb{E}_P[1]=1$, then we have the tighter representation: $D\_{\phi}(P||Q)   \coloneqq \operatorname*{sup}\_{{f}\in \mathcal{F}} \mathbb{E}\_{P} [f] - \operatorname*{min}\_{\lambda \in \mathbb{R}} \{ \lambda + \mathbb{E}\_{Q} [\phi^*(f-\lambda)] \}$
> This result, using the infimum over $\lambda$ in the spirit of Ruderman et al. [1], appears to have been independently proposed by Agrawal et al. [2]. As our proof in Theorem 4.2 revolves around the aforementioned equation, it is appropriate to refer to it as a tighter estimation of mutual information.
>
> [1] Ruderman et al. Tighter variational representations of f-divergences via restriction to probability measures. ICML2012
>
> [2] Agrawal et al. Optimal bounds between f-divergences and integral probability metrics. ICML2020
>
> **Q6: What's the meaning of "DRO bridges the gap"?**
>
> A6: While recent work such as MINE [2] and CPC [36] have demonstrated that InfoNCE is a lower bound for MI, their theoretical analyses still exhibit certain shortcomings [37]. MINE employs a critic in the dual variational form (I_DV) to establish a bound that is neither an upper nor a lower bound on mutual information. Simultaneously, CPC's proof relies on unnecessary approximations. Consequently, the challenge of bridging the gap between contrastive learning (CL) and mutual information estimation from a rigorous theoretical standpoint remains an open question. Fortunately, the power of DRO enables us to rigorously demonstrate the equivalence between CL and mutual information from a theoretical perspective.

---

> > ### Comment · Reviewer_C48x · 2023-08-14
> > **Response**
> >
> > Thank you for the clarifications. I specifically appreciate the clarification of M2: "a larger $f\_\theta$ value indicates a higher degree of similarity," which I misunderstood at the initial review phase. Given this, I increase the evaluation score from 4 to 5.

---

### Official Review · Reviewer_rLbF · 2023-07-06

**Soundness:** 3 good
**Presentation:** 3 good
**Contribution:** 3 good
**Rating:** 6
**Confidence:** 3

**Summary:**

This paper starts from the question why the naive form of CL is robust to sampling bias issue, resulting in empirical success in various areas. T this end, the authors first present the relationship between CL and DRO theoretically, where the DRO-constrained CL objective is conceptually equivalent to the objective of CL itself. They further show that the temperature in CL acts as a (inverse of ) robust radius in DRO constraint.  Based on these findings, the authors finally propose ADNCE, where the importance weight of negative samples follow Gaussian distribution. The proposed methods are validated via experiments on various domains.

**Strengths:**

- The paper reveals the relationship between CL and DRO in a more comprehensive way to address the bias-tolerant behavior of CL.
- Theoretical findings regarding \tau are validated empirically.
- The proposed framework and the ablative model based on the approximation of CL-DRO (Eq. 8) achieve meaningful performance improvement on the datasets.

**Weaknesses:**

- The design choice of ADNCE needs to be justified. Why such Gaussian-like weights are the most reasonable choice and can address the weakness mentioned in Sec. 5.1?
- Missing sensitivity analysis of \mu and \sigma in (12), comparing to those of \tau in InfoNCE. How much the results change as the hyperparameters vary?

**Questions:**

- Why controlling the variance of negative samples contributes to the success of CL?


**Limitations:**

Limitation has been addressed.

---

> ### Author Rebuttal · Authors · 2023-08-09
>
> # Response to Reviewer rLbF:
>
> Dear Reviewer,
>
> We appreciate your recognition of our contribution on the connection between CL and DRO. We also express our gratitude for your insightful inquiries regarding ADNCE. Below, we present responses to your comments:
>
> **Q1: Why such Gaussian-like weights are the most reasonable choice and can address the weakness mentioned ？**
>
> A1: Thanks for your insightful comment.  The primary motivation behind ADNCE is to mitigate the issues of over-conservatism and sensitivity to outliers. These limitations stem from the unreasonable worst-case distribution, which assigns excessive weights to the hardnest negative samples. Consequently, any weighting strategy capable of modulating the worst-case distribution to focus more on the informative region holds promising. In our ADNCE, we opted for Gaussian-like weights due to its flexibility and unimodal nature. However, alternative weighting strategies such as Gamma, Rayleigh or Chi-squared could also be employed. The following experiment demonstrates that these alternative weighting strategies can yield comparable results to Gaussian-like weights.
>
> |Weight Strategy| Probability density function|CIFAR10|
> |----------|----------|------|
> |Gamma-like| $w(x,m,n)=\frac{1}{\Gamma(m)n^m}x^{m-1}e^{-\frac{x}{n}}$ |91.74 |
> |Rayleigh-like| $w(x,m)=\frac{x}{m^2}e^{-\frac{x^2}{2m^2}}$ |91.73 |
> |Chi-squared-like| $w(x,m)=\frac{1}{2^{m/2}\Gamma{(m/2)}}x^{m/2-1}e^{-{x}/{2}}$ |91.99|
> |Gaussian-like (ADNCE)|$w(x,m,n)=\frac{1}{n\sqrt{2\pi}}e^{-\frac{1}{2} (\frac{x-m}{n})^2} $|91.88 |
>
> The table above presents a comparison of TOP-1 accuracy performance on the CIFAR10 dataset across different weight strategies. The parameters $m$ and $n$ are utilized to denote the parameters within their corresponding probability density function, while the variable $x$ represents a random variable. It is important to note that, due to the domain definition of some PDFs being $(0,+\infty)$, we need to set $x=prediction\\_score + 1$.
>
> As such, any proposal that enables adjusting the weight distribution centered on the main regions of probability density is a reasonable choice.
>
> **Q2: Missing sensitivity analysis of $\mu$ and $\sigma$？**
>
> A2: Much thanks for underlining this point. Here we presented sensitivity analyses of $\mu$ and $\sigma$ on cifar10, cifar100, and stl10 datasets. The results ar presented as follows:
>
> |    $\mu$     | 0.1  | 0.3  | 0.5  | 0.7  | 0.9  |
> |----------|------|------|------|------|------|
> | CIFAR10  | 91.3 | 91.66| 91.9 |91.77 |92.25 |
> | STL10   |87.84 |88.22 |87.56 |88.48 |88.45 |
> | CIFAR100 |69.34 |69.31 |68.70  |69.24 |68.95 |
>
>
>
> |    $\sigma$   | 0.2  | 0.4  | 0.6  | 0.8  |  1   | 1.5  |  2   |
> |------------|------|------|------|------|------|------|------|
> |  CIFAR10   | 90.07| 91.85| 92.02| 91.77| 91.72| 91.69| 91.94|
> |  STL10   | 86.54| 88.30 | 88.10 | 87.54| 88.95| 88.12| 88.40 |
> |  CIFAR100   | 67.38| 69.36| 69.52| 69.01| 68.70 | 69.24| 69.42|
>
> The above tables showcase the comparisons of TOP-1 accuracy performance on three datasets (CIFAR10, STL10, and CIFAR100) under different values of $\mu$ and $\sigma$. As can be seen, changing the parameters $\mu$ and $\sigma$ would impact the model performance, but not as dramatical as tuning the parameter $\tau$. (For example, on STL10 $\sigma$ from 1.0 to 0.2 just brings 2.7\% performance drops, while changing $\tau$ brings 12.6\% performance gap.) This outcome indicates that tuning $\mu$ and $\sigma$ is not a significant burden. In most scenarios, it may suffice to set $\sigma=1$, requiring only the tuning of $\mu$ within the range of 0.1 to 0.9 (can refer to Table 6, 7, 9 in Appendix B for more details).
>
>
> **Q3: Why controlling the variance of negative samples contributes to the success of CL？**
>
> A3: Controlling the variance of negative samples is crucial for the success of contrastive learning, for two reasons. On the one hand, by regulating the variance of predicted scores for negative samples, we simultaneously reduce the variance in model loss. This, in turn, enhances the generalization capability and reliability of the model. Analogous, there are numerous  studies [16, 26, 35] on the trade-off between bias and variance in model loss, demonstrating significant advancements in performance and robustness enhancement. On the other hand, variance control also manifests as rigorous hard-mining, as evidenced by the gradient expression presented below.
>
> $$
> \frac{d}{d\theta}(\mathbb{E}\_{Q\_0} [f\_\theta] + \frac{1}{2\tau} \mathbb{V}\_{Q\_0} [f\_\theta]) = \mathbb{E}\_{Q\_0} \left[\frac{d}{d\theta}(f_\theta)\right] +  \frac{1}{\tau}  \mathbb{E}\_{Q\_0} [(f\_\theta - \mathbb{E}\_{Q\_0} [f\_\theta]) \frac{d}{d\theta}(f\_\theta)]
> $$
>
> Higher weights are allocated to harder negatives having higher predicted scores, capturing the core idea of hard mining. Without variance control, weights lack distinction. This necessitates using a substantially larger set of negatives, further complicating training.

---

### Official Review · Reviewer_sD47 · 2023-07-06

**Soundness:** 3 good
**Presentation:** 3 good
**Contribution:** 3 good
**Rating:** 6
**Confidence:** 2

**Summary:**

The paper proposes a novel theoretical framework for understanding contrastive learning (CL) via distributionally robust optimization (DRO).
Under this framework, the paper derives that the InfoNCE loss can be interpreted as DRO with KL ball around the negative sampling distribution, and the paper leverages DRO insights to derive the optimal temperature parameter. Experiments are conducted which show that a novel modification of InfoNCE can lead to better sample complexities.

**Strengths:**

1. The framework that the paper proposes is quite elegant mathematically and explains the InfoNCE loss from a rigorous mathematical perspective, which endows us with many insights.
2. Using these insights, the paper proposes a new ADNCE loss that overcomes the issues of conservatism in DRO and shows some possibly promising experiments. (I am not an expert in CL so I cannot gauge how convincing these empirical improvements are).

**Weaknesses:**

1. The proposed approach introduces more hyperparameters, which define the weighting distribution. These hyperparameters can be a hassle to tune in practice, and it is unclear if the reported gains are simply from better hyperparameter tuning with these newly introduced hyperparameters.. It would be informative to show whether the new ADNCE is better for a large number of hyperparameters, or only a few.

Please also answer my questions.

**Questions:**

1. In Corollary 3.4, how is optimality measured? In other words, what objective does the setting of Eqn 6 optimize, and which assumptions are needed for it to hold?
2. Besides the math, is there any intuition on why DRO in the negative samples is what CL should strive for? In other words, why do we want to maximize the distance for all possible distributions of negative samples around the truth? (rather than the one we see data from). I think illustrating this intuition would be important to making this paper stronger.
2b. Does this DRO insight applies to other types of CL losses, besides InfoNCE?
3. What is Q^ideal? It is undefined before Line 154.
4. Can you please shed more light on what is hard-mining and how DRO gives us insights that hasn't been uncovered before? The paper seems to assume that readers know what hard-mining is, but I am not aware of this phenomenon for CL.
5. Can ADNCE also be explained with some modification of DRO? It would be nice to see if the new method is theoretically motivated as well, or simply a heuristic to avoid conservatism of DRO.


**Limitations:**

See weaknesses/questions.

---

> ### Author Rebuttal · Authors · 2023-08-09
>
> # Response to Reviewer sD47:
>
> Dear Reviewer,
>
> We greatly appreciate your acknowledgement of our contributions and your insightful comments. In what follows, we provide responses to the questions you have raised:
>
>
>
> **Q1: Concerning on the burden of hyperparameter tuning in ADNCE**
>
> A1: Considering the two limitations of InfoNCE, namely its over-conservatism and sensitivity to outliers, we introduce ADNCE that employs Gaussian-like weights with only two additional hyperparameters ($\mu$ and $\sigma$). In most scenarios, it may suffice to set $\sigma=1$, requiring only the tuning of $\mu$ within the range of 0.1 to 0.9 (can refer to Appendix B.1 and Table 6 in Appendix for more details). To further explore the model's sensitivity to the hyperparameter $\mu$, we have conducted additional experiments and got the following results:
>
>
> |    $\mu$     | 0.1  | 0.3  | 0.5  | 0.7  | 0.9  |
> |----------|------|------|------|------|------|
> | CIFAR10  | 91.3 | 91.66| 91.9 |91.77 |92.25 |
> | STL10   |87.84 |88.22 |87.56 |88.48 |88.45 |
> | CIFAR100 |69.34 |69.31 |68.70  |69.24 |68.95 |
>
>
> The table above presents a comparison of TOP-1 accuracy performance across three datasets at different values of $\mu$.
> As observed, the performance of our model does not exhibit high sensitivity to $\mu$. Even when $\mu$ is set to a less optimal value (for instance, $\mu=0.1$ in CIFAR10), our ADNCE still outperforms the original InfoNCE. This outcome underscores the efficacy of our ADNCE, and indicates that tuning $\mu$ and $\sigma$ is not a significant burden.
>
>
>
> **Q2: In Corollary 3.4, how is optimality measured?**
>
> A2: In this context, $\alpha^*$ denotes the optimal value of $\alpha$ that minimizes the initial formula as presented in equation (4). Specifically, $\alpha^*=\operatorname*{argmin}\_{\alpha\geq 0} \operatorname*{min}\_{\eta\_1} \operatorname*{max}\_{Q\in \mathbb{Q}} \{ \mathbb{E}\_{Q}[f\_\theta] -  \alpha [D\_{KL}(Q||Q\_0) -\eta] + \eta\_1 (\mathbb{E}\_{Q\_0} [\frac{Q}{Q\_0}]  - 1) \}$. Corollary 3.4 aims to analyze the relationships between $\alpha^*$ and $\eta$. Here, we simply apply a second-order Taylor expansion approximation without imposing any additional assumptions, which renders Corollary 3.4 both general and applicable across various domains. Appendix A.3 provides more comprehensive details.
>
>
>
> **Q3: Is there any intuition on why DRO in the negative samples is what CL should strive for?**
>
> A3: In constrastive learning, a uniform distribution is often employed to sample negative instances. However, this distribution is not optimal and may select instances with similar semantics, thereby leading to a potential issue of sampling bias. DRO empowers CL with a resilience to such sampling bias.
>
> DRO can be intuitively understood as a specific adversarial method: it introduces adversarial perturbations to the negative distribution, and the model is subsequently optimized to resist these perturbations. Through this mechanism, DRO entables CL to perform well accross various potential distributions, thereby endowing it with robustness against sampling bias.
>
>
> **Q4: What is Q^ideal?**
>
> A4: Thanks for underlining this problem. We will provide a comprehensive explanation of Q^ideal in the next revision. The term $Q^{ideal}$ denotes the ideal negative sampling distribution that selects instances with distinctly dissimilar semantics. It should be noted that in practical constrastive learning, the ideal $Q^{ideal}$ is not avaliable. As a substitute, the uniform distribution $Q\_0$ is utilized, which may sample instances with similar semantics, thereby introducing a so-called sampling bias. Theorem 3.3 is aimed at establishing theoretical bounded relationships between the model directly trained on the ideal $Q^{ideal}$ and the model trained on $Q\_0$ using InfoNCE.
>
> **Q5: What is hard-mining and how DRO gives us insights that hasn't been uncovered before?**
>
> A5: Hard-mining refers to a phenomenon that CL adaptively puts more weights on hard negative samples, which have higher prediction score and are challenging to distinguish. While this phenomenon has also been illuminated in recent studies either by examining the gradient magnitude [43] or through coordinate-wise optimization [45], our work presents the hard-mining property from a novel perspective and explicitly provides an expression of the weight.
>
> In addition to hard-mining, DRO unveils some attributes that have not been disclosed by recent work, including its robustness to sampling bias and the pivotal role of the temperature. Furthermore, the DRO perspective enhances our understanding of the limitations of InfoNCE, thereby inspiring us to develop a superior method ADNCE.
>
>
> **Q6: Can ADNCE be considered as a theoretical modification of DRO?**
>
> A6:  We are thankful for your insightful query. ADNCE is a heuristic approach designed to directly address the limitations of InfoNCE. While ADNCE may lack a formal theoretical foundation, this straightforward strategy indeed demonstrates superior performance across a variety of tasks. It would indeed be fascinating to explore the theoretical foundation of ADNCE or to devise a new method that incorporates a theoretical modification of DRO. We aspire to undertake such an endeavor in our future work.

---

### Official Review · Reviewer_CyLz · 2023-07-12

**Soundness:** 2 fair
**Presentation:** 2 fair
**Contribution:** 4 excellent
**Rating:** 6
**Confidence:** 4

**Summary:**

In this work the authors demonstrate a connection between contrastive learning, in particular InfoNCE, and distributionally robust optimization. In contrastive learning algorithms it is typical during training that samples which are similar are treated as being different, ie a negative pair, since contrastive learning typically does not include class labels. Intuitively this would be a hinder the ability of contrastive learning algorithms to learn useful representations. Interestingly this doesn't seem to happen in practice, choosing a temperature parameter correctly can cause standard CL algorithms to perform as good as, or better than, robust variants, in the presence of false negative pairs.

In this work it is theoretically proven that InfoNCE is naturally distributionally robust, which explains the phenomenon from the last paragraph. This is done by proposing a distributionally robust version of InfoNCE and demonstrating that the loss is equivalent to InfoNCE with scaling and shifting. This work goes on to analyze the effect and selection of the temperature parameter, giving further insight into the behavior of InfoNCE. It is demonstrated that the distributionally robust InfoNCE is an estiamtor of mutual information. This work then proposes ADNCE as a way to overcome shortcoings of InfoNCE, in particular its sensitivity to outliers, i.e. the hardest negative pairs. This is done by weighting samples in the loss so that the most outlying negative pairs have lower weight in the ending risk. It is experimentally demonstrated that this method brings improvements.

**Strengths:**

- The theoretical result of this work is useful, reasonably nontrivial, and regards a topic that is of very large interest to the ML community. Very good!
- ADNCE is an easily algorithm for a problem of signifiant interest, that brings consistent improvements. Also very good!

**Weaknesses:**

I spent a fair amount of time with this paper and I think its contributions are of high significance and importance, however I really feel that the exposition needs some improvement. I spent some time with the proofs and, while I was always able to decipher what was happening and found no errors, but they are written in a way that I found very hard to parse. While its far from the worst writing I have seen, there is quite a lot of the main text that was also difficult to precisely understand. I don't think the topicality here is so mathematically dense or advanced that this couldn't be written in a much clearer way. If this paper were very well written I would give this an 8, I think. Some example issues are listed below:

- Line 63: What is "the ultimate distribution"?
- Line 101 & Proof of Theorem 3.2 : it would be much clearer if $f_\theta$ was instead $f_\theta(x,y)$. This wouldn't take more space and it would be much easier to parse whats going in in the Proof of Theorem 3.2 if this were the case.
- Theorem 3.2: It should be made clear that $\alpha^*$ is a function of $\eta$, this seems quite important. One could simply write $\alpha^*(\eta)$.
- appx. line 10: I think $P$ in this line is supposed to be $Q$
- appx. (17):  I think its worth writing the original constrained optimization before applying strong duality.
- appx. (17): $\max_L$ is again a bit vague. I'm assuming whats actually happening is that we are optimizing over $Q$ in the $L$ definition.
- appx. line 21: "fine definition" is nonstandard, something like "always exists" or "is well-defined"
- appx. line 29: this should be an $\arg \min$. Text like "$\alpha^*$ represents the optimal value of ..." is needlessly confusing, just say "$\alpha^* = \arg \min \cdots$."
- Line 148, 165: Maybe I'm missing something, but I don't quite get what "optimal" means in these lines.
- Theorem 3.3: What is $Q^{ideal}$? In what sense is it "ideal"?

If this work used more clear and precise notation I think it would be great, right now it looks like a rough draft. As an example improvement, I would rewrite in appx (17) the fist term in the last line
$\mathbb{E}_{P_0} [f_\theta]$

as

$\mathbb{E}_{(X,Y)\sim P_0}[ f_\theta(X,Y)]$

Some other errors:
- "and graph" should be "and graphs"
- Give the definitions of acronyms, eg DCL HCL RPL
- There are many missing periods for sentences that end with equations, eg at the end of (1), (16),(17)
- Line 70: "downstrea" [sic]
- Page 3 footnote: sloppy ]

**Questions:**

I don't have any questions really, the paper just needs more polish

---

> ### Author Rebuttal · Authors · 2023-08-09
>
> # Response to Reviewer CyLz:
> Dear Reviewer,
>
> We sincerely appreciate your recognition of our work and deeply regret any confusion caused by typographical errors or unclear notations within our paper. Your detailed comments are highly valued. In the revised version, we commit to meticulously refining the paper in accordance with your feedback, rectifying any typographical errors or unsuitable notations, and providing precise explanations for each definition. Moreover, we will incorporate a comprehensive notation table to augment the clarity of our work. In the following, we provide responses to the questions you have raised:
>
> **Q1: What is "the ultimate distribution" in line 63?**
>
> A1: Here the ultimate distribution refers to the worst-case distribution $Q^*$ that the model is optimized on, formally defined as $Q^*=\operatorname*{argmax}\_{Q}  \mathbb{E}\_{Q}[f\_\theta]  \qquad s.t.  D\_{\phi}({Q}||{Q}\_0) \leq \eta $. From the Appendix A.6, $Q^*$ in InfoNCE can be written as $Q^* = Q\_0  \frac{\exp[f\_\theta / \tau]}{E\_{Q\_0} \exp[{f\_\theta / \tau}]}$. Our ADNCE reshape the $Q^*$ by incorporating Gaussian-like weights. In the forthcoming revision, we will replace the term "ultimate distribution" with "worst-case distribution" to enhance clarity.
>
>
> **Q2: What "optimal" means in lines 148, 165?**
>
> A2: In this context, the optimal $\alpha^*$ signifies the optimal value of $\alpha$ that minimizes the initial formula in equation (4). Specifically, $\alpha^*=\operatorname*{argmin}\_{\alpha\geq 0} \operatorname*{min}\_{\eta_1} \operatorname*{max}\_{Q\in \mathbb{Q}} \{ \mathbb{E}\_{Q}[f_\theta] -  \alpha [D\_{KL}(Q||Q\_0) -\eta] + \eta_1 (\mathbb{E}\_{Q_0} [\frac{Q}{Q\_0}]  - 1) \}$. By comparing equation (4) with InfoNCE, we deduce that the optimal $\alpha^*$ functions as the temperature $\tau$. Additonal, we employ the optimal $\alpha^*$ for analyses in Corollary 3.4 to gain insight into the nature of the temperature. We appreciate your insightful query and  will give more clear explanations of the optimal $\alpha^*$ in the next version.
>
>
> **Q3: What is $Q^{ideal}$ in theorem 3.3?**
>
> A3: The term $Q^{ideal}$ denotes the ideal negative sampling distribution that selects instances with distinctly dissimilar semantics. It should be noted that in practical constrastive learning, the ideal $Q^{ideal}$ is not avaliable. As a substitute, the uniform distribution $Q_0$ is utilized, which may sample instances with similar semantics, thereby introducing a so-called sampling bias. Theorem 3.3 demonstrates that InfoNCE is resilient to this sampling bias, establishes theoretical bounded relations between the model directly trained on the ideal $Q^{ideal}$ and the model trained on $Q\_0$ using InfoNCE.

---

> > ### Comment · Reviewer_CyLz · 2023-08-14
> >
> > The authors have presented a reasonable response to the questions the issues I mentioned. Its hard to know if the updated paper will overall be more clear without actually seeing it in its totality, but I can bump a point.

---

### Author Rebuttal · Authors · 2023-08-09

# Overall Rebuttal:
We thank all reviewers for taking the time to review our paper and for providing valuable and insightful feedback. We are delighted to see that our work has been recognized for its contributions and inspiration to the contrastive learning community, as mentioned by Reviewers $\color{red}{\text{CyLz}}$, $\color{green}{\text{sD47}}$, $\color{blue}{\text{C48x}}$, and $\color{orange}{\text{rLbF}}$.

We would like to express our gratitude to the reviewers for affirming the reliability and nontrivial nature of the theoretical framework we proposed from Distributionally Robust Optimization (DRO) perspective, as highlighted by Reviewers $\color{red}{\text{CyLz}}$, $\color{green}{\text{sD47}}$, and $\color{orange}{\text{rLbF}}$.

Regarding the introduction of the ADNCE loss, we are pleased to see that the reviewers found it both simple and significant, leading to effective results. We appreciate the positive feedback from Reviewers $\color{red}{\text{CyLz}}$, $\color{green}{\text{sD47}}$, and $\color{orange}{\text{rLbF}}$ in this regard.

We carefully considered the comments and suggestions provided by the reviewers, and we have addressed them point by point in our rebuttual. We believe and hope that our response adequately addresses the concerns raised by the reviewers.

Once again, we sincerely thank the reviewers for their valuable feedback, which has helped us improve the quality of our work.

---

### Decision · Program_Chairs · 2023-09-21

**Decision:**

Accept (poster)

**Comment:**

This paper draws connections between false negative resilience in contrastive representation learning with distributionally robust optimization. The reviewers agree that this is novel and interesting, and that the proposed practical algorithmic intervention (the ADNCE loss) is an appealing consequence.

There remain some concerns about clarity, which the authors are encouraged to address in the revision.